# AdaScape 1.0: a coupled modelling tool to investigate the links between tectonics, climate, and biodiversity

Esteban Acevedo-Trejos[1,*], Jean Braun[1,2], Katherine Kravitz[1], N. Alexia Raharinirina[3,4], and Benoît Bovy[1]

[1]Earth Surface Process Modelling, GFZ German Research Centre for Geoscience, Potsdam, Germany
[2]University of Potsdam, Potsdam, Germany
[3]Free University of Berlin, Berlin, Germany
[4]Zuse Institute Berlin, Berlin Germany

**Correspondence:** Esteban Acevedo-Trejos (esteban.acevedo-trejos@gfz-potsdam.de)

**Abstract.** The interplay between tectonics and climate is known to impact the evolution and distribution of lifeforms, leading to present-day patterns of biodiversity. Numerical models that integrate the co-evolution of life and landforms are ideal tools to investigate the causal links between these Earth system components. Here, we present a tool that couples an ecological-evolutionary model with a landscape evolution model (LEM). The former is based on the adaptive speciation of functional traits, where these traits can mediate ecological competition for resources, and includes dispersal and mutation processes. The latter is a computationally efficient LEM (FastScape) that predicts topographic relief based on the stream power law, hillslope diffusion, and orographic precipitation equations. We integrate these two models to illustrate the coupled behaviour between tectonic uplift and eco-evolutionary processes. Particularly, we investigate how changes in tectonic uplift rate and eco-evolutionary parameters (i.e. competition, dispersal, and mutation) influence speciation and thus the temporal and spatial patterns of biodiversity.

## 1 Introduction

Tectonic, climate, and evolutionary processes share an intrinsic co-evolutionary history (Lenton, 2004), which leaves salient patterns in the evolution and spatial distribution of lifeforms we observe today. For example, high biodiversity observed in mountain regions suggests a link between tectonics, climate, and the evolution of lifeforms (Fjeldsået al., 2012; Rahbek et al., 2019a, b). The Andean uplift led to increased topographic complexity and changes in climate (Hoorn et al., 2010), which prompted and sustained biodiversity in plants (Böhnert et al., 2019; Martínez et al., 2020; Pérez-Escobar et al., 2022), frogs and lizards (Boschman and Condamine, 2022), as well as fishes (Cassemiro et al., 2023). Similarly, a link between topographic complexity, associated climate changes, and high biodiversity have been proposed in highly diverse regions such as the Tibet-Himalaya-Hengduan region (Spicer , 2017; Ding et al., 2020) and Tropical Africa (Couvreur et al., 2021). However, we do not fully understand how tectonics and climate influence macroecological and macroevolutionary processes on large spatial and temporal scales. This requires a combination of approaches from multiple disciplines across the bio- and geosciences (Antonelli et al., 2018).

Understanding the large-scale temporal and spatial variation of lifeforms has been one of the central themes in various fields of ecology and evolution, such as macroecology (Brown and Maurer, 1989; McGill, 2019), historical biogeography (Wiens and Donoghue, 2004), macroevolution (Condamine et al., 2013), and more recently functional biogeography (Violle et al., 2014). Given the challenge of studying systems at such broad scales, these fields have utilised the use of simulation models to link ecological and evolutionary processes in a spatially-explicit context (Grimm et al., 2005; Gotelli et al., 2009; Connolly et al., 2017; Cabral et al., 2017). Currently, these types of models are known as "population-based spatially explicit Mechanistic Eco-Evolutionary Models" or MEEMs (Hagen, 2022) and prominent examples include Rangel et al. (2018) and Hagen et al. (2021a). In these models, the main emphasis is on how species interact and evolve, in a grid-based environment, where environmental fields (e.g. topography, temperature, and precipitation) are representations of past or present features computed a priori. These landscape representations are provided by other models, such as global paleo-elevation reconstructions (e.g. as in Hagen et al., 2021a) or after reanalysis of the output of other models (e.g. as in Rangel et al., 2018). These MEEM tools offer flexibility in the inputs and detail treatment of the ecological and evolutionary processes, however, they provide little control over the mechanisms that generate climate and landforms.

Simulating landforms requires considering tectonics, climate, and erosional processes, where particularly the latter can be mediated by organisms (Viles, 2020). Similarly, as in large-scale ecology, we can also investigate the processes leading to the formation of a particular topography using landscape evolution models (LEM) (Tucker and Hancock, 2010). But despite much progress in the field of biogeomorphology, there is a need for a new generation of LEMs, into a type of 'multipurpose modelling toolkit' as suggested by Viles (2020), that integrates landscape, ecological and evolutionary processes at large spatial and temporal scales (Badgley et al., 2017; Antonelli et al., 2018; Rahbek et al., 2019b). Nevertheless, such a toolkit should be simple enough to maintain generality, while capturing the relevant processes in macroecology, macroevolution, and geomorphology.

Here we present AdaScape, a coupled speciation and landscape evolution model conceived as a simple eco-evolutionary component built into an established LEM framework known as FastScape (Bovy, 2021a). The modelling framework is implemented in the programming language python and provides spatially-explicit environmental fields, e.g. topography and rainfall, while the AdaScape component contains routines to compute the adaptive speciation of individuals within the environment, which builds on the adaptive dynamics theoretical framework (Metz et al., 1996; Geritz et al., 1998; Dieckmann et al., 2004, 2007; McGill and Brown, 2007; Brännström et al., 2012). Organisms in such eco-evolutionary models are characterised by their traits and a fitness function that relates their trait values to local environmental conditions (McGill et al., 2006; Webb et al., 2010), thus the traits are not simply an input or a parameter of the model but a dynamic variable predicted by the model (Klausmeier et al., 2020). Below we describe in detail AdaScape and briefly FastScape, and provide a simple example to showcase the main features of the coupled modelling tool.

## 2 Model description

AdaScape is built on the simple eco-evolutionary model proposed by Irwin (2012), which describes the trait evolution of a group of individuals with processes related to environmental selection, mutation, and dispersal (Figure 1). We extend this model to include A) competition of a limiting resource influenced by individual traits, and B) more than one trait, which are related to environmental fields such as elevation and rainfall via simple linear function. The latter are environmental fields provided by a landscape evolution model (Figure 1 and section 2.5 below). Below we describe in detail the evolutionary and ecological components included in AdaScape as well as the taxon definition we use to reconstruct phylogenies and the basic elements needed to predict landscape evolution as included in FastScape.

### 2.1 Evolutionary components

Individuals $i$ are characterised by a vector of trait values $\mathbf{u}_i$ of length corresponding to the number of traits $k$. That is $\mathbf{u}_i = (u_{i,1}, u_{i,2}, \ldots, u_{i,k})^\mathsf{T}$. Environmental fitness (Figure 1) is given by a multivariate Gaussian function $f_i$, which reduces the fitness gain of the individual as its trait vector moves away from the optimal trait value vector $\mathbf{u}_0(z_i)$, where $\mathbf{u}_0(z_i) = (u_{0,1}(z_i), u_{0,2}(z_i), \ldots, u_{0,k}(z_i))^\mathsf{T}$ for a given local environmental condition $z_i$ as:

$$f_i(\mathbf{u}_i) = \exp\left(-\frac{1}{2}\left(\mathbf{u}_i - \mathbf{u}_0(z_i)\right)^\mathsf{T} \cdot \mathbf{\Sigma}^{-1} \cdot \left(\mathbf{u}_i - \mathbf{u}_0(z_i)\right)\right), \tag{1}$$

where $\mathbf{\Sigma}$ is the $k \times k$ matrix of parameters driving the fitness changes. Each diagonal element, $\sigma_{q,q}$, on the matrix $\mathbf{\Sigma}$, parametrises the fitness changes independently generated by the trait $u_{i,q}$ for all $q = 1, \ldots, k$. We assume that these diagonal elements are equal to $\sigma_f$ for all traits (Table 1). The parameter $\sigma_f$ determines the strength of the selection of an individual's trait value against its optimum and is here refer to as the environmental fitness variability. Each off-diagonal elements of $\mathbf{\Sigma}$, denoted $\sigma_{q,r}$, parametrises the fitness change jointly generated by the pair of traits at position $q$ and $r$ of the trait vector $\mathbf{u}_i$, for all $q, r = 1, \ldots, k$ such that $q \neq r$. We define $\sigma_{q,r} = \rho \sigma_{q,q} \sigma_{r,r}$ for all $q \neq r$ to model the interdependent effect of traits $u_{i,q}$ and $u_{i,r}$ have on fitness. The parameter $\rho$, thus, is here set equal to zero to simulate independent effects of all traits (Table 1).

The optimal trait value of the $q^{th}$ trait is given by a trait-environment relationship. Following Doebeli and Dieckmann (2003), we set this relationship to be linear:

$$u_{0,q}(z_i) = \alpha_z \cdot \left(Z_i(z_i) - \frac{1}{2}\right) + \frac{1}{2}, \forall q = 1, \ldots, k, \tag{2}$$

where $\alpha_z$ is a free parameter determining the slope of the relationship (Table 1), and $Z_i$ is the normalised environmental conditions experience by individual $i$. We use a normalised environmental field as these fields can change during the simulation. To facilitate the parametrisation, the ranges for an environmental field must be set up before the execution of the eco-evolutionary

model. Therefore, one can use the maximum $z_{max}$ and minimum $z_{min}$ ranges that each environmental field can reach during a simulation or the known ranges for a particular taxon or clade. The full expression for $Z_i(z_i)$ is given by:

$$Z_i(z_i) = \frac{z_i - z_{min}}{z_{max} - z_{min}}.\tag{3}$$

Mutation is the second evolutionary process we consider (Figure 1), which is described as an intergenerational stochastic variation of the trait values. The mutation process is thus model as a stochastic process occuring at probability, $p_m$ at every generation time. This process is typically simulated using a simple Monte Carlo sampling algorithm. The algorithm draws a random number from a uniform distribution $\mathcal{U}(0,1)$ and compare it with a mutation probability $p_m$ (Table 1); if the drawn number is less than $p_m$ then the descendant of the individual $i$ can mutate. Now a mutation is characterise by a new a trait value taken from the Gaussian distribution $\mathcal{N}(u_{i,q}, \sigma_m)$ centred at the ancestor trait value $u_{i,q}$ and with a mutation variability $\sigma_m$ (Table 1).

The mutation process is rather more complicated in nature as we simplify it here and can be defined in its broadest sense as a heritable change in the genome, which can be measured as the number of base pair substitutions per generation and is positively related to the genome size of the organism (Lynch, 2010). Changes in the genotype could lead to phenotypic variation that may be positively or negatively influenced by natural selection, therefore mutations could be silent, advantageous or deleterious (Bromham, 2016). Our model simplifies this genetic–phenotypic complexity by assuming that a change in the genotype (via mutation) directly corresponds to a change in the phenotype (trait), the latter of which is subject to selection and transmitted by uniparentally inherited markers (in the absence of sexual selection) as in Irwin (2012).

The third evolutionary process we consider is dispersal (Figure 1), where the new location of an individual $i$ is randomly sampled around the position of each individual $l_{i,x}$ and $l_{i,y}$, along the X and Y axes using separated Gaussian distribution $\mathcal{N}(l_{i,\bullet}, \sigma_d)$, where $\bullet$ is a place holder for x or y. Individuals' dispersal ability is influenced by their dispersal variability $\sigma_d$ (Table 1), which is here considered as a free parameter. In other words, dispersal describes the random movement of individuals and their traits through the landscape, where the new location of individuals at $t+1$ depends on the location of individuals at time $t$ and $\sigma_d$.

## 2.2 Ecological component

The main ecological interaction we consider in AdaScape is via a trait-mediated competition (Figure 1). In the original model of Irwin (2012) all the individuals, $n_{all}$, in the local neighbourhood were assumed to compete for a local resource. The latter can sustain a given number of individuals, or local carrying capacity $K$ (Table 1). The extent of the local neighbourhood is defined by a radius $r$ (Table 1) and is centred at each individual location. We modify this assumption by accounting not only for all individuals in the local neighbourhood but specifically for those individuals with similar trait values to the centred individual

$i$. For this, we introduce a term to account for the effective number of individuals $n_{eff,i}$ similarly to Doebeli and Dieckmann (2003). The expression accounting for the trait-mediated competition is given by:

$$n_{eff,i} = \sum_{j \in D_{i,r}} \exp\left(-\frac{\Delta_u^2(i,j)}{2\sigma_u^2}\right), \tag{4}$$

where $D_{i,r}$ is the local neighbourhood of radius $r$ of individual $i$, $\Delta_u(i,j)$ is the trait distance between individual $i$ and its neighbour $j$, and $\sigma_u$ is trait-distance variability or width (Table 1). The local neighbourhood is the area around each individual

centred at its location $l_{i,\bullet}$ and with an extend determined by $r$, and can be thought of as the area where local competition for resources take place. The parameter, $\sigma_u$, dictates the strength of the competition among individuals with similar trait values. Hence if $\sigma_u \geq 1$, all individuals in the local neighbourhood regardless of their trait values are assumed to compete for the same resource. However, if $\sigma_u < 1$ only those individuals with similar trait values to the focus individual $i$ are counted and thus assumed to compete for the same resource. In other words, the similarity in trait values is determined by how small $\sigma_u$ is.

Hereafter we consider two contrasting cases of this process that we define without ($\sigma_u = 2$) and with ($\sigma_u = 0.2$) trait-mediated competition.

## 2.3   Implementation details of the eco-evolutionary model

The model is implemented as an individual-based, spatially-explicit model in python. A simulation is initialised with a given number of individuals allocated randomly or at a particular range in a continuous 2D space. The traits for each individual are

drawn from a uniform distribution, where the minimum and maximum range is between 0 and 1. In all simulations hereafter we start with a monomorphic population, i.e. all individuals descent from the same ancestor and share similar trait-values. After initialisation, the fitness for each individual is evaluated following equation 1. Then we compute the number of offspring $n_{off,i}$ for each individual $i$ following Irwin (2012) using:

$$n_{off,i} = \frac{K}{n_{eff,i}} f_i, \tag{5}$$

where the $\frac{K}{n_{eff,i}}$ is the density dependent reproductive factor. After the number of offspring has been determined, the new individuals are generated, mutated, and dispersed. The two latter are implemented as stochastic processes as explained in the previous section. This model thus assumes that a generation is completed after all individuals have been updated, therefore, generations do not overlap.

## 2.4   Taxon definition

We define a taxon as a group of individuals sharing similar trait values and common ancestry (sensu Pontarp et al., 2012). We implement this by using a spectral clustering algorithm (von Luxburg, 2007), which examines individuals per ancestor group and assigns them to new taxon groups based on the clustering of their trait values at each time step. To each of these

clusters we assigned a new taxon-id $\in \mathbb{N} = \{0, 1, 2, 3, \ldots, \infty\}$, this taxon-id at time $t$ will become the ancestor-id at time $t+1$. For example, if one assumes that all individuals share a common ancestor and all of them have very similar trait values (a monomorphic population), then branching does not occur and all individuals will be assigned to a single taxon-id (e.g. equal to 1 if the ancestor-id is equal to 0). Conversely, if the clustering algorithm found two distinct trait clusters, then branching occurs and the individuals are clustered in two new taxon groups with different ids (e.g. 1 and 2 if the ancestor-id is equal to 0), with this we are avoiding polytomies by considering only binary splits. At the next time step and after the calculation of the eco-evolutionary processes (see details above), the previous taxon-id becomes the ancestor-id, and we apply the spectral clustering algorithm again using the new ancestor group. In our simulations, we restricted the division of taxa to a maximum of two to avoid the excessive occurrence of branching. Additionally, to add more interpretability to our taxa-clusters, we assume that the similarity between a pair of individuals is 0 when their trait distance is greater than a threshold $\tau$ (Table 1). This means that smaller values of $\tau$ instruct the algorithm to prioritise the grouping of the corresponding individuals but ignore the trait-distant information between all individuals that do not satisfy the threshold criteria. The choice of taxon threshold $\tau$, thus, depends on a trade-off between high-similarity grouping and valuation of trait-distance information, and here we chose a quite low taxon threshold to prioritise the grouping of highly similar individuals. This allows us to reconstruct lineages of the extant and extinct taxon to their last common ancestor and compute various phylogenetic metrics on synthetic phylogenetic trees. In Fig. 2 we illustrate how this algorithm works starting with a monomorphic population of individuals at time $t_0$, which then diversifies into two taxa at time $t_1$ and then further diverges into three taxa at time $t_2$.

This taxon definition is primarily based on the divergence of individuals' traits since they are the main dynamical variables affecting selection, mutation, and competition. However, the spectral clustering algorithm can be modified to accommodate other variables (e.g. geographical location of individuals or time since the last branching point). Furthermore, the taxon definition is independent of the eco-evolutionary processes of the model, thus changing the taxon definition would not affect the model behaviour.

## 2.5 Landscape evolution component

The environment where organisms adapt is here defined at a landscape scale and can consider common landforms such as mountains, plateaus, stream valleys, basins, and floodplains, among others. These landforms and their evolution can be reproduced using a Landscape Evolution Model (LEM), which in essence describes the changes in topography $h$ by the competition of processes that shape Earth's surface such as uplift and erosion (Whipple, 2004; Tucker and Hancock, 2010) as:

$$\frac{dh}{dt} = U - I + H, \tag{6}$$

where the first term $U$ is the uplift rate ($m/yr$, table 1), the second term $I$ is the river incision or stream power law (SPL) (Lague, 2014) and the last term accounts for hillslope processes. The river incision is the main erosional process of landforms and in simple terms describes how a flow of water cuts through bedrock and can be given by:

$$I = k_f \cdot \nu^a \cdot A^a \cdot S^b, \tag{7}$$

where $k_f$ is the constant of erodability ($m^{1-2a}/yr$, table 1), $\nu$ is precipitation rate (scaled by a reference rate) $A$ is the drainage area ($m^2$) and $S$ is the slope of the terrain. The latter two enter the SPL as power functions with, respectively, exponents $a$ and $b$. As the rivers cut through the valleys they create slopes that are thus subject to processes such as soil creep, landslides, and debris flows. Hence, the last term in equation 6 describes the transport of such material from hilltops to lowlands, also known as hillslope processes, which is determined by a constant transport coefficient or diffusivity $k_d$ ($m^2/yr$, table 1) and the curvature of the terrain ($m/m^2$) as:

$$H = k_d \cdot \nabla^2 \cdot h. \tag{8}$$

Modelling fluvial incision by stream power equation requires finding the numerical solution of a partial differential equation with linear and nonlinear slopes, which posed stability, accuracy, and speed constraints (Tucker and Hancock, 2010). However, one can overcome these issues by using FastScape (Braun and Willett, 2013), which is an efficient algorithm to compute the discharge at each node in an orderly manner following the steepest descent of the water flow to the base level in the landscape. This algorithm has been implemented in the FastScape framework (Bovy, 2021a) together with various other processes affecting landforms, such as orographic precipitation (Smith and Barstad, 2004), sediment transport and deposition by rivers (Yuan et al., 2019) among many other tectonic, climatic and erosional processes. While elevation is the main output of the LEM, this environmental field could be used as proxy for temperature. This would require further assumptions for example, that temperature decreases with elevation around 6.5°C/km (Minder et al., 2010) and a given baseline temperature at sea level, which could be constant or change over time and taken from climatic paleo-reconstructions.

In particular, orographic precipitation is a crucial environmental process that affects the availability of water for biota and influences the surface processes briefly mentioned before. FastScape computes rainfall fields using the linear orographic precipitation model proposed by Smith and Barstad (2004), which takes into account the topography, direction, and speed of the wind to predict the spatial distribution of rainfall by solving the advection equations that integrate cloud water density and rain/snow density using a Fourier transformation into the two horizontal directions (Smith and Barstad, 2004).

Explaining the details of all the processes included in FastScape go beyond the scope of this paper and we refer the reader to the documentation of the framework and related publications (Whipple, 2004; Smith and Barstad, 2004; Tucker and Hancock, 2010; Braun and Willett, 2013; Lague, 2014; Yuan et al., 2019; Bovy, 2021a). We use the SPL with hillslope processes and orographic precipitation to demonstrate how the distribution and evolution of taxa respond to dynamic changes in the topography and precipitation. To model precipitation and landscape evolution, we select values of the uplift rate $U$, the constant of erodability $k_f$, the transport coefficient $k_d$, a background precipitation rate $P_0$, wind speed $w_s$, and wind direction $w_d$. A description of the parameters and the values used in the examples below can be found in table 1.

Lastly, to connect the eco-evo model with LEM the local environmental condition $z_i$ is equal to the elevation $h$ or the orographic precipitation $\nu$ fields as provided by FastScape at the position of the individual $i$ at every time step. Therefore these

environmental fields form the basis to compute the optimal trait value (Equation 2) that each individual compares to quantify its fitness (Equation 1).

## 3 Examples

### 3.1 Evolutionary branching along a linear environmental gradient

Our first example considers the evolution of one trait representing the adaptation of individuals to topographic elevation, which is here simply termed as "trait elevation". The simple eco-evolutionary model produces patterns of evolutionary branching under fewer than 500 time steps along a continuous environmental gradient. Where evolutionary branching is here referred to as the split of a population with an average trait value into two populations with a progressive widening gap between their average trait values. The emergence of evolutionary branching along a continuous environmental gradient is a well-known

phenomenon captured in eco-evolutionary models that build on the adaptive dynamics theoretical framework (Metz et al., 1996; Geritz et al., 1998; Dieckmann et al., 2004; Klausmeier et al., 2020) and exemplified by the seminal work of Doebeli and Dieckmann (2003).

In Fig. 3 we show two such results for a case without trait-mediated competition ($\sigma_u = 2$, Fig. 3 A-E) and with trait-mediated competition ($\sigma_u = 0.2$, Fig. 3 F-J) in a simple 2D environment, where the environmental gradient (e.g. elevation)

linearly increases along the X-coordinate. The other parameters for these simulations are $\sigma_f = 0.2$, $\mu_m = 0.005$, $\sigma_m = 0.05$, and $\sigma_d = 30$. We set the local carrying capacity $K$ to 50 for the case without trait-mediated competition and $K = 35$ for the case with trait-mediated competition. This parameterisation leads to a roughly equal saturation in the total number of individuals in the two scenarios (Figure 3A and F) in contrast to an equal $K$ that would predict higher individual abundances in the case of trait-mediated competition. Changes in total abundance (via changes in carrying capacity) are known to affect

the number of taxa in this type of eco-evolutionary models that use a similar taxon definition (Pontarp and Wiens, 2017). Therefore, to minimise density-dependent effects on taxon richness we assure that both cases reach similar total abundances ($\approx 400$ individuals, Fig. 3 A and F) by reducing local carrying capacity, but see Appendix A for a sensitivity analysis of the effects of selected parameters on the maximum abundance of individuals.

Both simulations show branching, but when competition among individuals with similar trait values is strengthened, further

branching is promoted (cf. Fig. 3 C and H). Figures 3-B and -G show the phylogenetic tree and trait distribution of the extant taxa without and with trait-mediated competition (Figure 3 C and H). The phylogenetic reconstruction using our proposed taxon definition (Figure 3 E and J) resembles the pattern of population trait values over time both in terms of the number of branches and the trait distributions of each branch (Figure 3 C-D and H-I). However, this method would also separate taxa even if branches on a population level cannot be distinguished (i.e. when organisms occupy all trait space). Furthermore, the taxon

method can create phylogenies when organisms are described by more than one trait.

## 3.2 Biodiversity patterns in static vs. dynamic landscapes

Our second example considers the evolution of two traits representing the adaptation of species to topographic elevation and to orographic precipitation, here termed as "trait elevation" and "trait precipitation", respectively. This experiment shows how different biodiversity patterns can emerge from the interaction of eco-evolutionary and Earth surface processes using AdaScape. For this we consider two contrasting environmental histories that produce the same final mountain belt: A) a **static landscape** where the topography and precipitation do not change over time (i.e. no uplift or erosional processes) and B) a **dynamic landscape** where both topography and orographic precipitation change as a function of uplift over time. In Fig. 4 we show the predicted topography and precipitation for these two model set-ups in an idealised landscape of 100 km by 100 km and for a total simulation time of about 10 Myr with time steps of 10 kyr. The resulting landform consists of a mountain belt with a main drainage divide in the middle of the model domain (Figure 4 A-C and D-F), which reaches a maximum height of ≈5 km. This high topography creates an orographic barrier to the wind that moves in a south-to-north direction. Hence producing the typical high precipitation on the windward slope of the mountain belt and a rain shadow with drier conditions on the leeward slope of the mountain belt (Figure 4 H-J and K-M). The simulations for static and dynamic landscapes reach equivalent mean, maximum, and minimum values of elevation and precipitation.

We parameterise two eco-evolutionary models: one without ($\sigma_u = 2$, Eq. 4) and another with ($\sigma_u = 0.2$, Eq. 4) trait-mediated competition, which we then run in the static and dynamic landscapes (Figure 4). We start all simulations with a monomorphic population of around 100 individuals (Figure 5 - A, D, G, and J) where all individuals have similar trait values set to 0.25 for the trait associated with elevation (Figure 5 - B, E, H, and K) and 0.75 for the trait associated with precipitation (Figure 5 - C, F, I, and L). This represents an initial population composed of individuals adapted to lowlands and high precipitation. To avoid large differences in the fitness values of the initial populations, we set the individuals to start at specific locations either in the southern portion or at random locations in the landscape for the static or dynamic landscape conditions, respectively. We assume that the relationship between the optimal trait value and the environmental field is positive for both traits (i.e. $\alpha_z = 0.95$ Eq. 2). The traits are considered to be independent ($\rho = 0$, Eq. 1) and the value for environmental fitness variability is set as a strong selection for traits around the optimal trait values ($\sigma_f = 0.2$, Eq. 1). Mutation probability, $p_m$, is set to 0.005 and the mutation variability $\sigma_m$ to 0.05, which introduces a small intergenerational trait variability. We parameterise dispersal variability, $\sigma_d$, to 10 km. Local carrying capacity, $K$ (Eq. 5), is parameterised to 50 (without trait-mediated competition) and 25 (with trait-mediated competition) individuals where the radius of the local neighbourhood $r$ is set to 20 km.

For the coupled execution of the eco-evolutionary model into the LEM we have to assume that one generation time is equal to one time step of LEM. This of course can lead to unrealistic generation times that exceeds the average lifespan of organisms. Therefore, careful consideration of the parameters is required when AdaScape is coupled with FastScape since one generation would then represents the temporal aggregation of numerous real generations (see section 5 for discussion on the scaling limitations).

These two contrasting environmental conditions lead to distinct temporal patterns for simulations without and with trait-mediated competition. As the simulation progresses, the number of individuals increases until reaching similar total abundances

of around 350 individuals (Figure 5 - A, D, G, and J) and different diversification patterns emerge (Figure 5 - B-C, E-F, H-I, and K-L), in particular, between a static and dynamic landscape (cf. Figure 5 B-C against E-F and H-I against K-L). Under a static landscape, the evolutionary branching occurs sooner than under dynamic landscape conditions, because in the latter the individuals first are selected for narrow environmental ranges, which then extend towards the end of the simulation. The environmental conditions progressively increase during the first 2 Myrs of the simulation (Figure 4), which leads to the narrowly observed trait variability. Between 2 and 6 Myr the environmental gradients extend into broader precipitation and elevation ranges (Figure 4) and consequently lead to an increase in trait variability. After 6 Myrs the environmental fields reach their maximum extent (Figure 4), with little trait variability until the end of the simulation (Figure 5).

The reconstructed phylogenetic history for the taxa at the end of the simulation summarises the emergent diversity patterns of the four example simulations (Figure 6). We observed that simulations without trait-mediated competition lead to lower taxon richness (i.e. 3 and 6 taxa for static and dynamic landscape) compared to simulations with trait-mediated competition (i.e. 25 and 22 for static and dynamic landscape). We can distinguish a division between those clades from mostly wet-adapted taxa in the South and mostly dry-adapted taxa in the North (cf. red and blue coloured lineages in Figure 6). Such a division between the northern and the southern clades seems to coincide with the increase in the range of the environmental gradients, in particular, under dynamic landscape conditions (cf. Figure 6 C and D with Figure 4 G and N). This suggests a relationship between the rate of change in environmental conditions and the response in the build-up of biodiversity.

### 3.3 Effects of uplift, mutation, and dispersal variability on biodiversity

To investigate how the build-up of biodiversity is influenced by the rate of change in environmental conditions and eco-evolutionary processes, we varied three parameters, namely uplift rate, dispersal variability, and mutation variability (Table 1). In figure 7 and 8, we quantified how these changes affect when biodiversity reaches its maximum in terms of number of lineages through time (LTT). For the two competition cases, we tested three different values of uplift, dispersal and mutation centrered around the paramaterisation used in the example in section 3.2 (Figure 5 and 6, Table 1). For, each parameter set, we repeated the simulation 10 times for a total of 60 simulations. In figure 7, we normalised each LTT by the maximum number of lineages reached on each simulation.

We observed that as the mountain is uplifted faster, the maximum number of lineages is reached earlier (Figure 7A and B). Conversely, the peak in the number of lineages is delayed as the rate of uplift slows (Figure 7A and B). Changes in uplift rate similarly affect simulations with and without trait-mediated competition (cf. Fig. 7A with B). However, in absolute terms, the simulations with trait-mediated competition lead to a higher number of lineages (Figure 8), while the overall patterns remain similar to the normalised values (Figure 7).

The eco-evolutionary processes also show differences in the timing with respect to the values tested and between cases of competition. On the one hand, a low dispersal variability leads to a slower build-up of diversity compared to intermediate and higher values (Figure 7C and D). However, as time progresses the intermediate and high dispersal cases tend to reach their maximum earlier, while the simulations with low dispersal values continue to increase (Figure 7C and D). This is reflected in the highest number of lineages, in absolute terms, reached for the low dispersal with trait-mediated competition case (Figure

8C and D). On the other hand, increasing mutation variability causes a faster build-up of diversity, while the contrary occurs when the values of mutation variability decrease (Figure 7E and F). In absolute terms, an increase in mutation variability tends to increase the number of lineages, with the highest values predicted with trait-mediated competition case (Figure 8E and F).

## 4   Comparison with similar modelling approaches

Our eco-evolutionary implementation is built on the model proposed by Irwin (2012), where he showed how phylogeographic structure (i.e. historical geographic distribution of clades) emerges along an environmental gradient as selection, dispersal, mutation, and population size vary. Irwin's model is similar to earlier eco-evolutionary models, such as Doebeli and Dieckmann (2003), which described adaptive speciation patterns (or evolutionary branching of a trait) along environmental gradients. Doebeli and Dieckmann (2003) showed the range of parameters where branching is facilitated and the importance of ecological processes. Particularly, they demonstrated that when competition strength is smaller than the selection strength branching is promoted (Doebeli and Dieckmann, 2003). Albeit in Irwin's original model, competition for resources was not considered, we show here that including such ecological process facilitates speciation (Figures 2 and 6). Both works also show how an increase in dispersal leads to well-mixed and spatially unstructured populations (Doebeli and Dieckmann, 2003; Irwin, 2012), which will thus dampen the number of lineages. A pattern we also capture with our model (Figure 7 and 8).

Contemporary to Irwin's work, Pontarp et al. (2012) proposed an eco-evolutionary model also inspired by Doebeli and Dieckmann (2003), but using a different fitness generating function and reconstructing taxa and phylogenies based on the similarity of trait values and shared common ancestry. The latter helps Pontarp et al. (2012) to extend the traditional application of these types of eco-evolutionary models from population to community. They have used this model to show how A) phylogenetic structure emerges in communities competing for resources (Pontarp et al., 2012), b) mode of speciation (from sympatric to allopatric) can change continuously (even during a single radiation event) depending on local to regional conditions and dispersal capacity of organisms (Pontarp et al., 2015), and C) how richness patterns along gradients depend on the carrying capacity, diversification rates, and time-for-speciation (Pontarp and Wiens, 2017). We adopted a similar approach to that of Pontarp and colleagues to define taxa, which allows us to broaden the applicability of the model and reconstruct idealised phylogenies that can be compared with time-calibrated phylogenies.

Another characteristic of the works of Irwin (2012) and Pontarp et al. (2012) is that they divide the environmental gradient into discrete habitats, while Doebeli and Dieckmann (2003) use a spatially continuous and linear environmental gradient. Haller et al. (2013), building on the work of Doebeli and Dieckmann (2003), tested the effects that various spatially complex environments (i.e. linear gradients, nonlinear gradients, and spatially continuous patches) have on branching. They found that an intermediate level of environmental heterogeneity promotes branching, and they suggested using metrics of their realised environments to compare with observations in real landscapes. In addition, Doebeli and Dieckmann (2003) demonstrated, early on, the impact of the relationship between the slope of the environmental gradient with dispersal by revealing that evolutionary branching is facilitated at intermediate environmental gradients once dispersal is below a critical level. Here by coupling our

adaptive speciation model to a landscape evolution model, we not only produce a more realistic landscape but show the impact of considering an environmental gradient that changes over time (i.e. dynamic landscape).

A recent tool named the *gen3sis* engine can simulate ecological and evolutionary processes in paleo-geographies that are changed at discrete time-steps (Hagen et al., 2021a). This tool was used to investigate the effects that plate tectonics and paleoclimate reconstructions have on macroecological patterns of diversity, such as the latitudinal diversity gradient (Hagen et al., 2021a), and pantropical diversity disparity (Hagen et al., 2021b). This engine offers a great wealth of outcomes (e.g. species distributions, phylogeny, ecological traits) that can seamlessly be compared with empirical observations. Other similar models that mainly focused on the ecological and evolutionary aspects have been proposed earlier (Rangel et al., 2018), in what is known as "population-based spatially explicit mechanistic eco-evolutionary models" or MEEMs for short (Hagen, 2022). Our model differs from MEEMs in that we follow an individual-based (IBM or agent-based ABM) modelling approach, which in comparison to population-based models offers greater flexibility into the processes considered and how the organisms interact among themselves and with the environment (Levin, 1998; Railsback , 2001; DeAngelis and Mooij, 20005; Grimm et al., 2005). IBMs thus account for low-level variability that can be scaled up to higher hierarchical levels, producing emergent properties that cannot be predicted by the properties of individuals or their interactions with the environment alone (Levin, 1998; Railsback , 2001; DeAngelis and Mooij, 20005; Grimm et al., 2005). Nevertheless, accounting for individual-level variability, particularly as observed in nature, can be impractical and computationally demanding. Hence, population-based models, such as MEEMs are a more computationally efficient option (Hagen, 2022). In addition, MEEMs such as Rangel et al. (2018) and Hagen et al. (2021a) do not compute the landscape dynamics but use a priori calculated environmental fields. This approach allows them a more flexible and efficient way to upscale the computations. However, this comes at the cost of not controlling the relevant processes that lead to the building of a landform or climate as in our coupled eco-evolutionary and landscape evolution model.

Recent efforts to couple eco-evolutionary models with a LEM, such as BioSlant (Stokes and Perron, 2020), present a more promising venue to explore the link between tectonics, climate, and biodiversity. However, Stokes and Perron (2020) implemented a different eco-evolutionary model that captures mainly allopatric speciation. This model is based on an earlier version of a neutral metapopulation model (Muneepeerakul et al., 2007) where speciation is not linked to functional traits and the organisms only move along river networks. Hence, a species or taxon in this type of model is a static definition. Since mobility is limited to river networks the main application is to investigate the effects of river reorganisation on biodiversity. Our approach circumvents these limitations through the continuous interplay between organisms traits and their environment.

## 5   Limitations of our modelling approach

Any model at its best is a surrogate of nature that we can use to test our understanding of a system. We decided to build on established theoretical frameworks to study the coupled eco-evolutionary dynamics (Metz et al., 1996; Geritz et al., 1998; Dieckmann et al., 2004; McGill and Brown, 2007; Klausmeier et al., 2020) and landscape evolution (Whipple, 2004; Tucker and Hancock, 2010; Lague, 2014). These theoretical frameworks have had many applications over past decades with more detailed

descriptions of processes as we have shown here, we thus use only their essential processes to illustrate how bio- and geo-components of the Earth system can be coupled into a single modelling framework while keeping the number of parameters and processes to a minimum. While it is appealing to integrate ecological and evolutionary processes into a landscape evolution model several caveats related to the scaling of such processes needs to be taken into consideration. Three types of scaling problems are recognized in ecological models, namely pre-model (i.e. (dis)aggregation of values used as input of models), in-model (i.e. procedures related to the simplification of a model), and post-model (i.e. scaling procedures applied to the output of models) issues (Fritsch et al., 2020). Here we focus on the pre-model and in-model scaling issues, particularly those related to maintaining computational efficiency while keeping the ability to detect emergent speciation patterns (Fritsch et al., 2020).

Coupling these types of models comes at the expense of increased computational cost. To prevent this we develop our simple eco-evolutionary model on a very efficient algorithm to solve the stream power law (Braun and Willett, 2013) and its implementation using xarray-simlab infrastructure (Bovy et al., 2021b). This implementation known as FastScape provides a series of libraries to efficiently compute and parallelise simulations (Bovy, 2021a). Hence our model executes in the order of minutes with computational costs exponentially increasing as the number of time steps and the maximum number of individuals (local carrying capacity) increases (Figure 9). For all simulations, the spatial extent consists of an area of 100 km by 100 km, which is divided in a regular grid of 100 by 100 points. The simulations were executed for 10 Myr with variable time steps, as explained in figure 9. Albeit we run these simulations in a 176-core (Intel Xeon 2.10Ghz) Linux cluster, the execution of FastScape and AdaScape do not require any high-performance computing facilities and can be executed in any modern desktop or laptop computer where Python can be installed with similar performance as shown in figure 9.

When AdaScape is coupled with FastScape, the time between generations is defined as the time step of the LEM (i.e. 10 kyr for the example simulations shown in section 3.2). Therefore, a generation in AdaScape would represent the temporal aggregation of numerous real generations. In this context, a scaling of the eco-evolutionary parameters related to mutation ($p_m$, $\sigma_m$) and dispersal ($\sigma_d$) must be considered. The simplest way is to scale the mutation and dispersal parameters by the square root of the number of real generations in a LEM time step. Therefore, the parameter $p$ used in a given simulation with a time step $\Delta t$ should be scaled for comparison with measured values, $p'$, by the following relationship: $p' \sim p \cdot \sqrt{t_G/\Delta t}$, where $t_G$ is the average lifespan of a generation, $p$ is one of the mutation or dispersal parameters, and $p'$ is the new scaled parameter. Caveats of this scaling are that the parameter value is not directly constrained by observations and that obviously, this parameter should not be too large in a simulation to avoid, for example, that dispersal exceeds the length of the simulated area or that all individuals mutate and vary the trait value broadly in a single time step. Particularly, the latter goes back to the assumptions of earlier formulations of adaptive dynamics, stating that the range of validity of these types of eco-evolutionary models stay as long as mutation processes were rare and the trait variability with respect to their ancestor was small, to assure that evolutionary dynamics were slower than changes in population densities (Abrams, 2001; Klausmeier et al., 2020).

In addition, keeping a tractable number of individuals during the simulations can also be a challenge. Further developments can use scaling-up procedures, for example, the so-called super-individuals approach (Scheffer et al., 1995) to be able to represent even higher abundances of individuals. This would require limiting the number of individuals to a predefine maximum and minimum number, where each of these super individuals accounts for the properties of several others.

## 6    Competition as driver of diversity

Competition for resources is an important ecological process (Tilman, 1982; Chesson, 2000), that can lead to the divergence of traits and consequently promote biodiversity (Pfennig and Pfennig, 2009). Examples of both interspecific (e.g.: Grant and Grant, 2006; Grainger et al., 2021) and intraspecific (e.g.: Bolnick, 2001; Calsbeek and Cox, 2010) competition, are known to leave an imprint on the traits under selection. Therefore, ecological processes have the potential to alter the outcome of evolution. Increasing interest in the past decades have been in documenting cases where ecological dynamics and evolutionary

dynamics show reciprocal interactions (Fussmann et al., 2007; Schoener , 2011; Govaert et al., 2019), thus leading to a recurring call to integrate the distinct disciplines of ecology and evolution, as recently point out by Loreau et al. (2023). This becomes particularly relevant knowing that both ecological and evolutionary dynamics can operate at the same pace (Fussmann et al., 2007; Schoener , 2011; Govaert et al., 2019) and can be influenced by rapid changes of the environment, for example as the climate changes (Parmesan, 2006; Loreau et al., 2023). Hence, the difficulty of understanding the tangled relationships between

the biotic and abiotic environment with the ecological and evolutionary responses of organisms. Our model, although aiming at capturing the essential eco-evolutionary processes, simplifies much of the organism-organism and organism-environment feedback. Nevertheless, the results support the general view that competition is an important process that promotes the build-up of taxon diversity.

## 7    Conclusions

There is a great appeal for numerical tools that look to integrate various components of the Earth system such as tectonics, climate, and biodiversity, albeit such tools are not common (Antonelli et al., 2018). Some of the existing tools focus on the eco-evolutionary components, leaving behind Earth's surface processes and climate, while others do not factor in eco-evolutionary dynamics and trait-environment relationships. Here we introduce our coupling of a simple eco-evolutionary model into a very efficient landscape evolution model *FastScape*, which offers great potential to explore the links between the three main

components of the Earth system into a single modelling framework and at manageable computational costs. This allows, as we show here, to perform a large number of simulations and consider ensemble properties rather than those emerging from single simulations, the details of which can be highly dependent on the initial conditions or the stochastic nature of the evolution equations representing mutation and dispersal. At the moment AdaScape mainly considers the effect that the environment has on the biota, however, feedbacks between these components can be further investigated. For example, by linking the effects

that organisms such as plants have on the hydrological cycle, which are known to dampen the discharge variability (Rossi et al., 2016) and therefore the erosional efficiency of rivers where erosional thresholds exist (Deal et al., 2018; Braun and Deal, 2023). This would require to investigate and test relationships between organismal traits with erosional processes in the landscape, which offer a potential venue to further integrate various components of the Earth system into a single modelling framework.

*Code and data availability.* All routines of AdaScape and FastScape are openly accessible by GitHub (https://github.com/fastscape-lem/adascape) and Zenodo (https://doi.org/10.5281/zenodo.7794374). Also we include the procedures to reproduce all the results presented here.

## Appendix A: Sensitivity on the maximum number of individuals to changes in selected parameters

Due to stochastic and nonlinear relationships of the processes, as well as our definition of the local neighbourhood in the eco-evolutionary model, it is difficult to know a priori how the different processes will interact to produce a maximum number of individuals. Therefore, we perform a sensitivity analysis on the maximum number of individuals, $N_{max}$, to changes in the parameters $\sigma_f$, $\sigma_u$, $\sigma_d$ in relation to changes in radius $r$ and local carrying capacity $K$. We then tested 10 values for the following ranges of $\sigma_f$=[0.2, 2], $\sigma_u$=[0.2, 2], and $\sigma_d$=[10, 100] in relation to changes in 10 values in the range of $K$=[25, 75] and $r$=[25, 75]. We performed 100 simulations for each pair-wise set of parameters for a total of 700 simulations (Figure A1). We observed that the maximum number of individuals will be reached as $r$ decreases and $K$ increases. Also, the maximum number of individuals when $K$ increases and $\sigma_u$ and $\sigma_d$ decreases or when $\sigma_f$ increases. Similarly, the highest $N_{max}$ is reached when $\sigma_u$ and $\sigma_d$ decreases or when $\sigma_f$ increases, in combination with a decrease in the radius of the local neighbourhood $r$.

## Appendix B: Trait-mediated competition, optimal trait-environment relationship and environmental heterogeneity

The effects of competition in our model depend on the relationship between the environmental field and optimal trait, as well as on the heterogeneity in the environment. To better illustrate this, in Figure B1 we performed a model run as in Example 3.1 for the case with competition ($\sigma_u$), where we set the environment – optimal trait relationship ($\alpha$) from the default value 0.95 to 0, the latter meaning that there is no relationship between the trait and the environmental field. We then run this model setup under an environment with a linear environmental gradient (as in Figure 3) and under no environmental gradient, i.e. a constant environmental field centred at a middle elevation. We observe no build-up of taxon diversity when there is no environment–optimal trait relationship ($\alpha = 0$) without (Figure B1 A-D) and with (Figure B1 E-H) linear environmental gradient, and when $\alpha > 0$ with a linear environmental gradient (Figure B1 I-L). Trait-mediated competition thus only promotes diversity when a trait environmental relationship exists and when the simulation occurs in a heterogeneous environment (as in Figure 3 F-J).

*Author contributions.* EA-T extended and developed the eco-evo model, performed numerical simulations and analysis, and wrote first draft. JB contributed to the conceptualisation of the model, the development and implementation of the algorithm as well as the design of the numerical experiments. KK developed and tested the initial eco-evolutionary models, consulted on the extended model development. NAR helped to implement the eco-evo model. BB consulted on the extended model development and contributed to the implementation of the algorithm into FastScape. All authors provided edits on the manuscript and have agreed to its submission for publication

*Competing interests.* We declare no competing interests.

*Acknowledgements.* We thank the German Research Foundation (DFG) that financed EA-T and JB via Collaborative Research Centre 1211 "Earth evolution at the dry limit" (project number 268236062) and NAR under Germany's Excellence Strategy – The Berlin Mathematics Research Center MATH+ (EXC-2046).

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

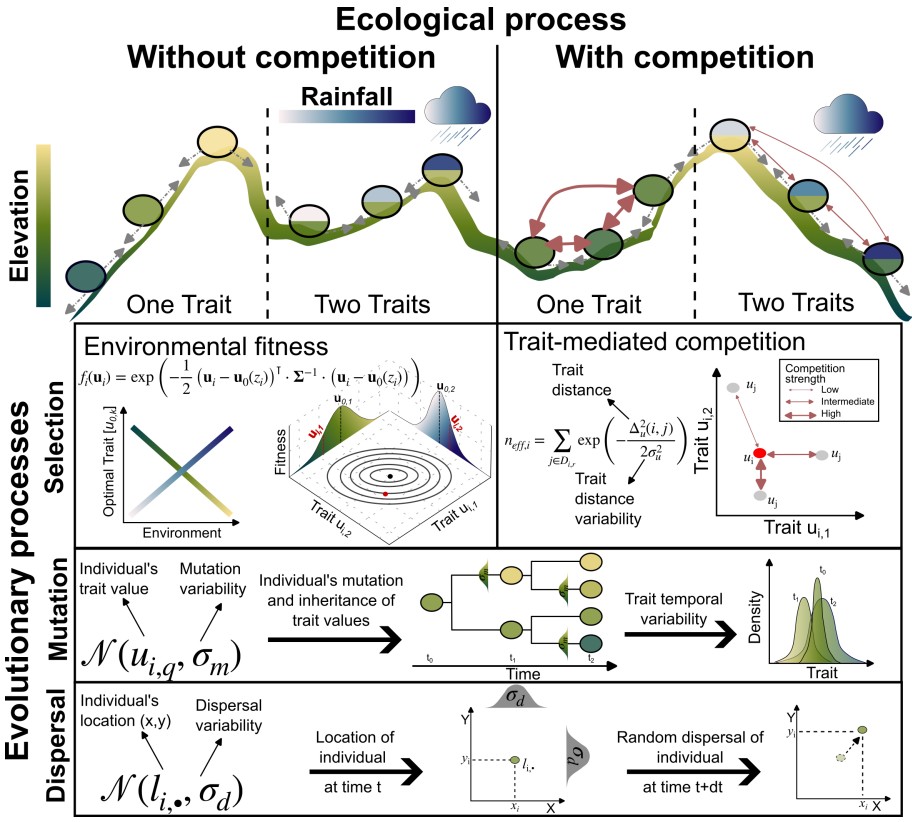

**Figure 1.** Schematic illustrating the main ecological and evolutionary processes included in AdaScape for two traits (i.e, $k = 2$) representing the adaptation of the population to topographic elevation and to rainfall, respectively. The eco-evolutionary model is a modified version of the one proposed by Irwin (2012). Selection is one of the crucial eco-evolutionary processes included, which is here determined by the environmental fitness or how suitable the trait values of an individual $\mathbf{u}_i = (u_{i,1}, u_{i,2})^\intercal$ compare to the optimal trait value $\mathbf{u}_0(z_i) = (u_{0,1}, u_{0,2})^\intercal$ for a given local environmental condition, $z_i$. The other main selection process is trait-mediated competition, which determines how many individuals with similar trait values to the focal individual $i$ are competing for the same local resource. We also include mutation and dispersal as stochastic processes that depend, respectively, on the trait $(\mathbf{u}_{i,q}, \forall q = 1, \ldots, k)$ and location $\mathbf{l}_{i,\bullet} = (l_{i,x}, l_{i,y})$ of the individual $i$ and the parameters that control the variability or width of the trait value ($\sigma_m$) and location ($\sigma_d$) the offspring will inherit or disperse to.

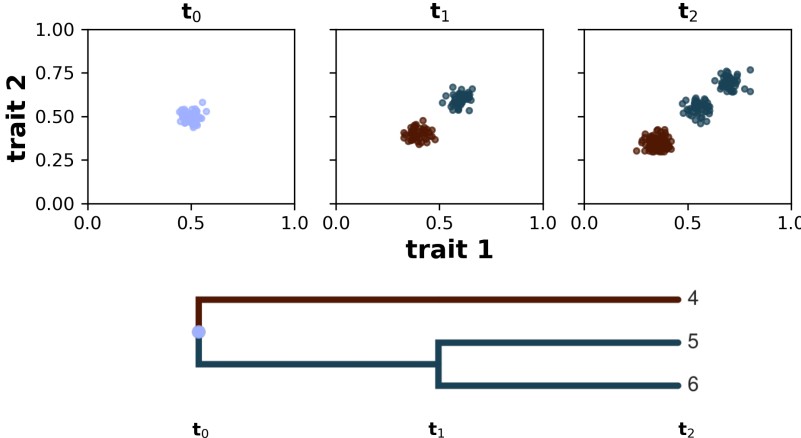

**Figure 2.** Taxon definition implemented in AdaScape by using a spectral clustering algorithm. The algorithm groups individuals according to their common ancestry and similarity of trait values. In the three subplots in panel A we show the distribution of individuals in trait space along the axis $trait\ 1$ and $trait\ 2$ at three-time steps: $t_0$, $t_1$, and $t_2$. Below the subplots we show the phylogenetic reconstruction of the identified taxa, starting from the last common ancestor with a taxon-id equal to 0 all the way to taxa 4, 5 and 6 at time $t_2$.

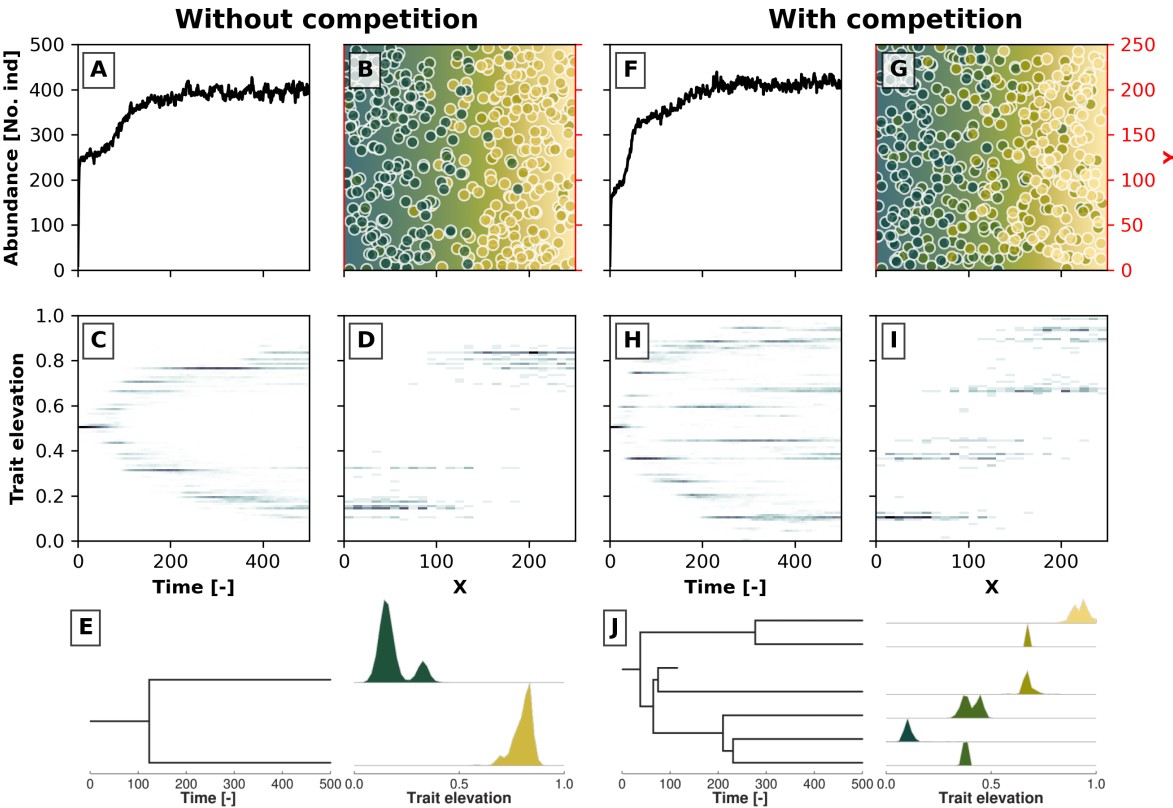

**Figure 3.** Example simulations along a 2D gradient showing the effects without (A-E) and with (F-J) trait-mediated competition have on evolutionary branching patterns. A and F show the temporal changes in the number of individuals. C and H show the trait distribution over time in a 2D histogram, where the darker colour marks a higher number of individuals with a given trait value at a particular time. B and G show the spatial distribution of individuals in the 2D environment at the last time step, where the blue colour reflects lower trait values and the yellow colour reflects high trait values. The coloured dots represent individuals with their corresponding trait values. D and I show the distribution of trait values along the X coordinate at the last time step. In panels E and J we reconstruct the phylogenetic tree for, respectively, the simulations without and with trait-mediated competition. At each extant branch in the phylogenetic tree, we plotted the trait distribution of that particular taxon.

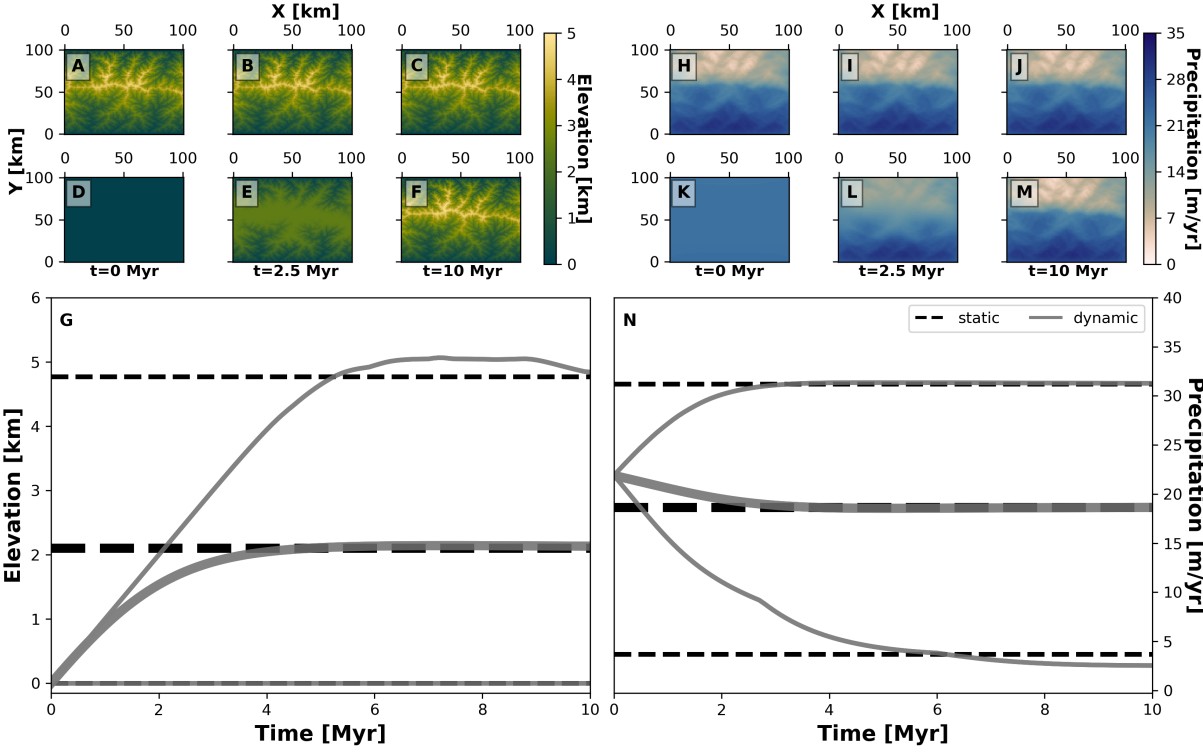

**Figure 4.** Spatial and temporal patterns of environmental fields under static and dynamic landscape conditions. We consider two main environmental fields elevation (A-G) and precipitation (H-N). For each, we consider two types of environmental histories one for a static landscape where conditions are constant (A-C, H-J and black dashed lines in G and N) and another for a dynamic landscape for which conditions vary through the simulation (D-F, K-M, and grey solid lines in G and N). The thick line marks the mean and the thinner lines are the minimum and maximum of each environmental field. To produce these environmental conditions we consider the following parametrisation of our landscape evolution model (Table 1): $U = 0\,m/yr$ (static) or $0.001\,m/yr$ (dynamic), $k_f = 0\,m^{1-2a}/yr$ (static) or $2.8 \cdot 10^{-6}\,m^{1-2a}/yr$ (dynamic), $k_d = 0\,m^2/yr$ (static) or $0.01\,m^2/yr$ (dynamic), $a = 0.4$, $b = 1$, $P_0 = 22\,m/yr$, $w_s = 15\,m/s$, $w_d = 0°$.

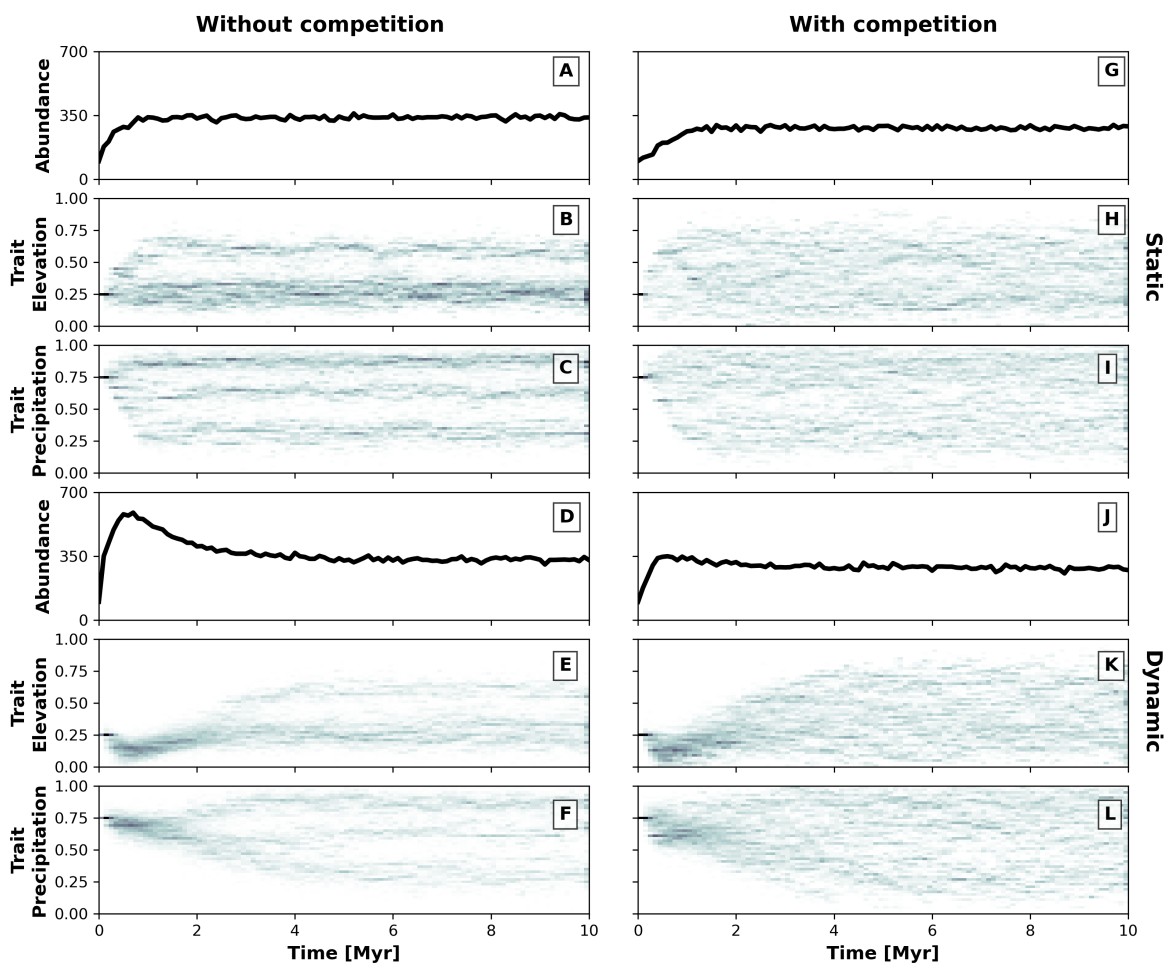

**Figure 5.** Temporal dynamics of the eco-evolutionary model without (A-F) and with (G-L) trait-mediated competition and each for a static (A-C and G-I) and dynamic (D-F and J-L) landscape. Where A, D, G and J show the number of individuals over time. B-C, E-F, H-I, and K-L present the trait distribution over time. We use 2-dimensional histograms for presenting the temporal distributions of traits (B-C, E-F, H-I, and K-L), where the darker colouration highlights the higher frequency of individuals with a particular trait value.

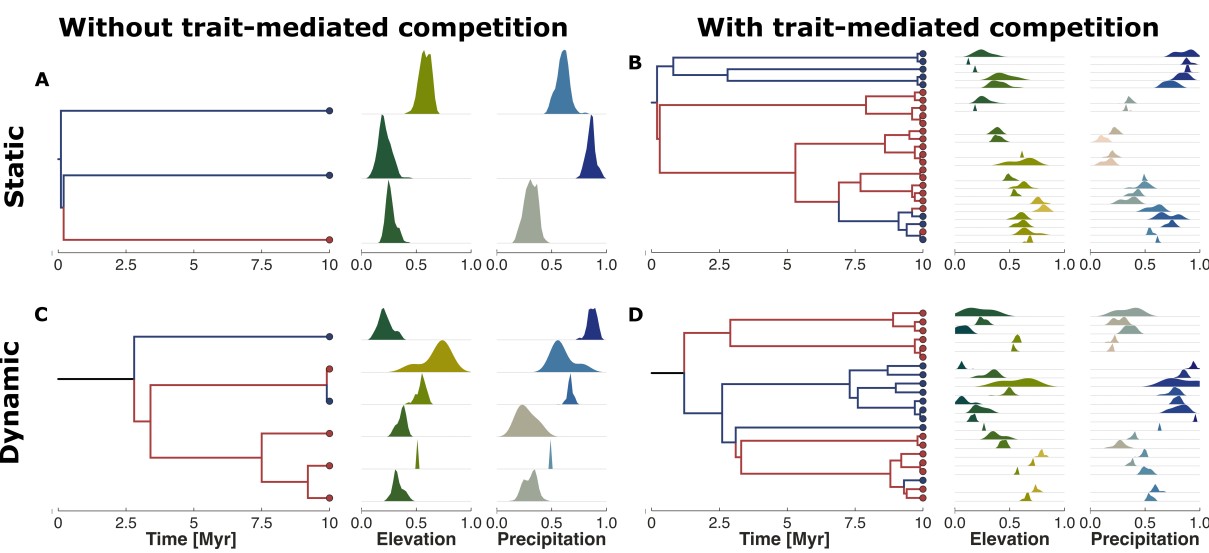

**Figure 6.** Phylogenetic reconstruction of the extant taxa at the end of the simulation. The subplots represent our four examples without (A and C) or with (B and D) trait-mediated competition and under environmental conditions of a static (A and B) or dynamic (C-D) landscape (Figure 4). The red and blue colour circles in the tips of the phylogenetic trees highlight, the upper half (north) and lower half (south) average location of the taxa along the Y-coordinate. We similarly marked the branches with the same colour coding to better distinguish the relationships between the North dry-adapted (red) and South wet-adapted (blue) clades. The density plots on the right of each tree show the trait distribution for each taxa and for traits associated to Elevation and Precipitation.

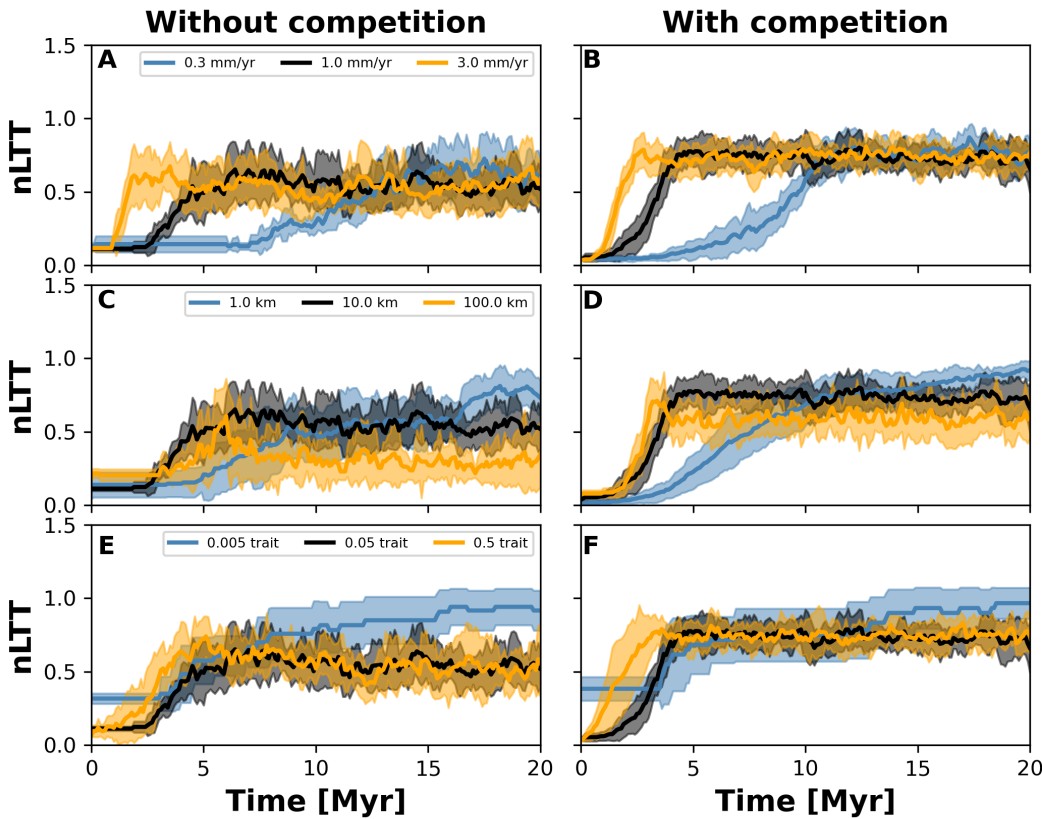

**Figure 7.** Normalised lineages through time (nLTT) plot summarising numerical experiments, where we investigate the effects of uplift rate (A-B), dispersal variability (C-D), and mutation variability (E-F). The results for each treatment are the mean (solid line) and standard deviation (shaded area) of 10 replicates with different random seeds. To facilitate the comparisons among treatments we normalised the number of lineages to the maximum number that each single replicate reached. We also double the simulation time to 20 Myr but kept time step 10 kyr (cf. Fig. 6) to assure that both the landscape (in the case of different uplift rates) and number of taxa reach an equilibrium.

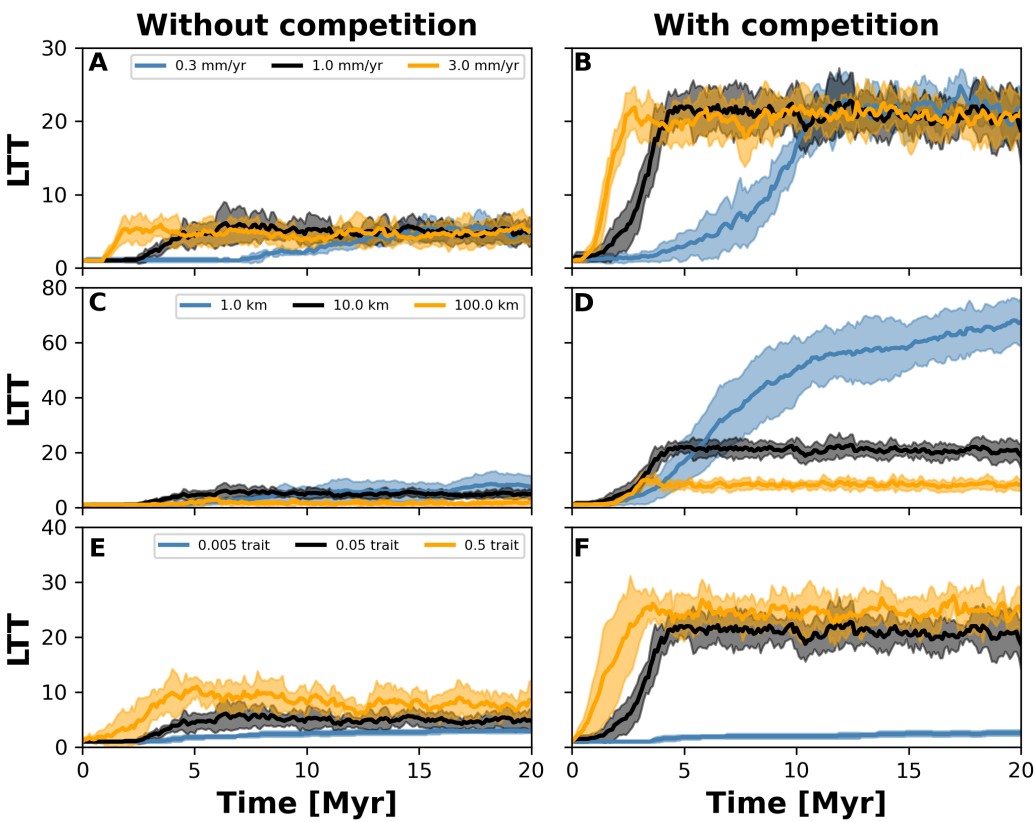

**Figure 8.** Number lineages through time (LTT) plot summarising numerical experiments, where we investigate the effects of uplift rate (A-B), dispersal variability (C-D), and mutation variability (E-F). The results for each treatment are the mean (solid line) and standard deviation (shaded area) of 10 replicates with different random seeds. Results based on same observations as in Fig. 7.

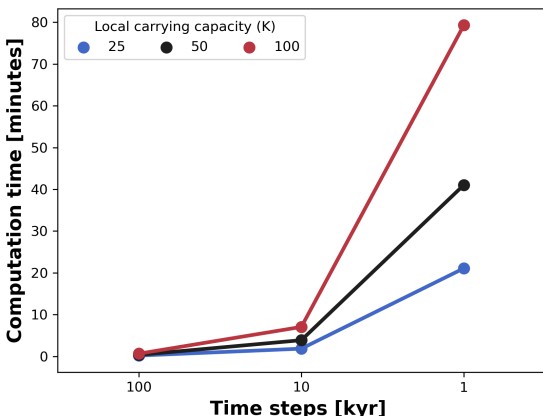

**Figure 9.** Average computation time for a single AdaScape run. We measured the average time variation between 7 runs setup without trait-mediated competition as shown in examples (Figures 5 and 6). We then manipulated the local carrying capacity and the length of the time steps in the model. For all simulations, the spatial extent consists of an area of 100 km by 100 km, which is divided in a regular grid of 100 by 100 points. The simulations are executed for 10 Myr with variable timesteps in a 176-core (Intel Xeon 2.10Ghz) Linux cluster.

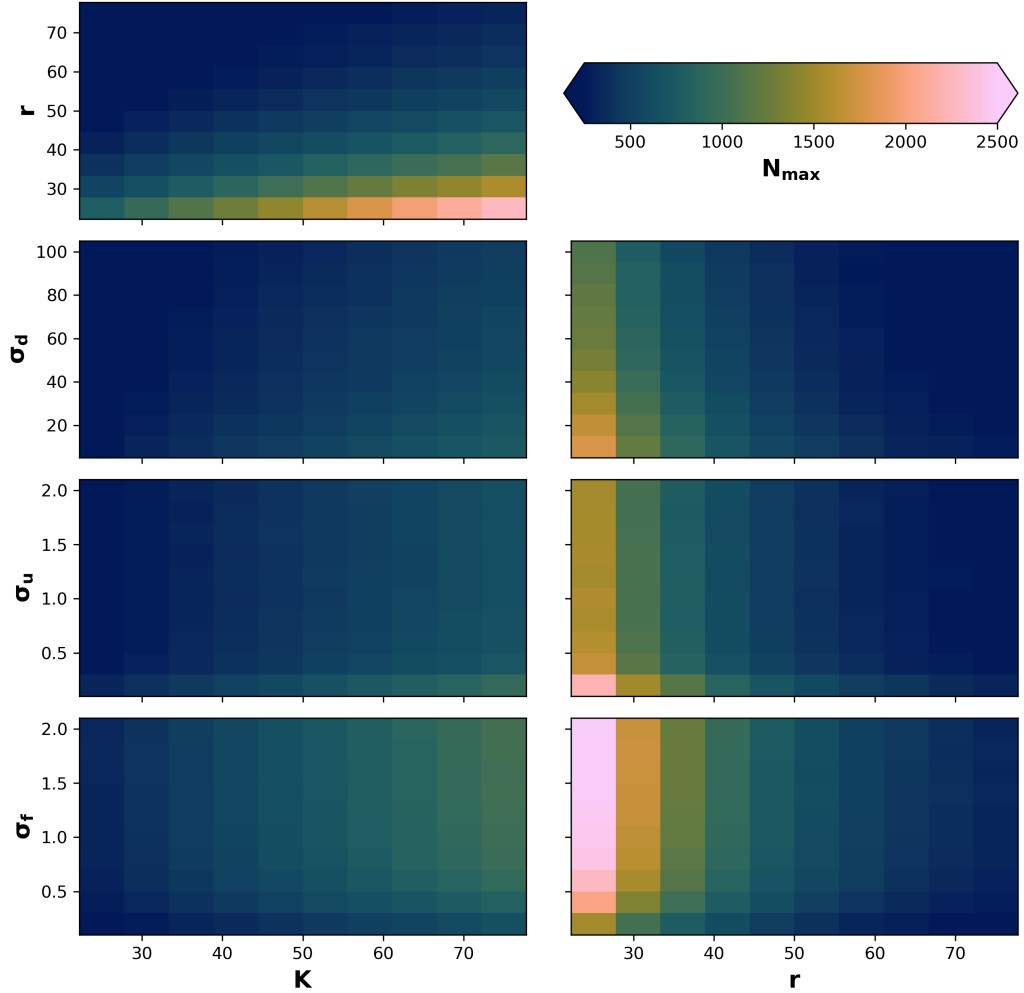

**Figure A1.** The maximum number of individuals, $N_{max}$, reached during a simulation given a set of parameters. The model results are calculated using the Example 3.1 setup for a linear environmental gradient. The default parameter values are: $r$=30, $K$=50, $\sigma_f$=0.2, $\sigma_u$=2, and $\sigma_d$=30.

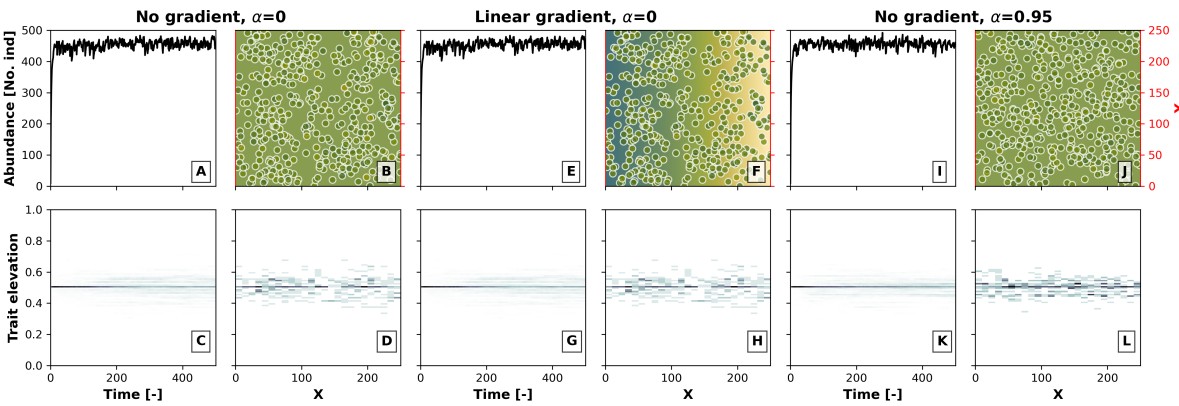

**Figure B1.** Effects of the environment - optimal trait relationship ($\alpha$) and competition on the build-up of taxon diversity. The numerical experiments show the results of simulations with 1) no environmental gradient (same elevation throughout the landscape) and no environment - trait relationship $\alpha$ (A-D), 2) linear environmental gradient (as in Figure 3) and no environment - trait relationship (E-H), and 3) no environmental gradient and a positive environment - trait relationship $\alpha = 0.95$. The other parameter values are set as in Figure 3 for the case with the trait-mediated competition.

**Table 1.** Description of parameters in the adaptive speciation model, together with a selection of parameters we vary to reconstruct the topography and rainfall patterns. We use the default values for all other parameters in the landscape evolution and orographic precipitation model.

| Description | Symbol (Units) | Values |
|---|---|---|
| Environmental fitness variability | $\sigma_f$ (trait) | 0.2 |
| Mutation probability | $p_m$ (-) | 0.005 |
| Mutation variability | $\sigma_m$ (trait) | [0.005, 0.05, 0.5] |
| Dispersal variability | $\sigma_d$ (km) | [1, 10, 100] |
| Trait competition variability | $\sigma_u$ (trait) | [0.2, 2] |
| Radius of local neighbourhood | $r$ (km) | 20 |
| Local carrying capacity | $K$ (No. ind.) | [25, 50] |
| Correlation coefficient among traits | $\rho$ (-) | 0.0 |
| Slope optimal trait - environmental field | $\alpha_z$ (trait) | 0.95 |
| Taxon threshold | $\tau$ (-) | 0.075 |
| Uplift rate | $U$ (m/yr) | $[0, 3 \cdot 10^{-4}, 1 \cdot 10^{-3}, 3 \cdot 10^{-3}\ ]$ |
| Erodability coefficient | $k_f$ (m$^{1-2a}$/yr) | $[8.4 \cdot 10^{-7}, 2.8 \cdot 10^{-6}, 8.4 \cdot 10^{-6}]$ |
| Transport coefficient | $k_d$ (m$^2$/yr) | [0, 0.01] |
| Precipitation base | $P_0$ (m / yr) | 22 |
| Wind speed | $w_s$ (m / s) | 15 |
| Wind direction | $w_d$ (degrees) | 0 |