# Peer review of "AdaScape 1.0: a coupled modelling tool to investigate the links between tectonics, climate, and biodiversity"

_Geoscientific Model Development, 2023_

## Author Comment (AC1)

We are grateful with the reviewer's helpful comments, which have greatly improved the manuscript. Below we provided a point-by-point response to the suggestions raised by the reviewers (smaller *italics* font) together with our response (larger **bold** font) and modified text (larger blue font).

**RC1**

*General*

*This manuscript describes a new coupled Landscape Evolution Model (LEM) and eco-evolutionary model called AdaScape. The LEM is FastScape, a well-known model that simulates river erosion and hillslope erosion an uplifting the landscape. The authors additionally incorporate orographic precipitation into the LEM component of the model. They then couple FastScape with an eco-evolutionary model in which individuals experience trait evolution in response to selection, mutation, dispersal, and competition. The authors explore elevation and precipitation as organismal traits in the models presented in the paper. The results show that trait-mediated competition has a large influence on the number of taxa. The landscape dynamics also influence the number taxa, but most significantly the rate of uplift influences the timing of lineage diversification. I think that AdaScape will be a very useful addition to the growing field of coupled Earth-Life science models. The paper is generally well-written, and the model clearly communicated. I think the manuscript will be improved with a deeper consideration and discussion of some topics that I detail below. There are some minor grammatical and wording issues throughout that I've noted in the section "Technical Corrections."*

**Thank you for the overall positive appraisal and corrections to our manuscript.**

*Specific Comments*

*Taxon Definition: If I understand the taxon definition correctly, some degree of trait divergence is necessary for speciation to be recognized in the model. However, speciation can occur (and as recent work has shown, may most frequently occur) in the absence of trait divergence. For example, Anderson and Weir (2022) compared sister species pairs of birds, mammals, and amphibians and find that the majority evolve under similar selective pressures (and show little trait divergence). I wonder if the authors have considered other definitions of taxon groups that would not require trait divergence, perhaps geographic distance and/or time. You could imagine, for example, two populations becoming isolated in lowlands across a mountain range – they may undergo allopatric speciation because of a geographic barrier even though their traits may not diverge substantially. While the current taxon definition may be useful for some specific questions, it may not be useful for replicating real-world patterns where allopatric speciation seems to be common.*

**The theoretical frameworks that Irwin's (2012) and consequently our model build on are known as adaptive dynamics (Metz et al., 1996; Geritz et al., 1998; Dieckmann et al., 2004), which is related to early work on quantitative genetics and evolutionary game theory (Abrams et al., 1993; McGill and Brown, 2007). More recently this broad theoretical framework can be better described as trait-based eco-evolutionary theory (Klausmeier et al., 2020). These theoretical frameworks share the commonality that the traits of an organism are not simply an input parameter but a dynamic variable predicted by the model (Klausmeier et al., 2020). Hence, we favour a taxon definition based on the divergence of individuals' traits over geographic variables since in our model the former (traits) are the main dynamical variables affecting selection, mutation, and competition. Geographically distinct populations (i.e. groups of individuals with distinct traits located at different areas of the landscape) could emerge in our model as a by-product of the frequency-dependent selection to different environmental conditions but taxon definition in our model is not directly related to geographic isolation. Hence, to your point about considering other taxon**

**definitions, you are correct that this definition would not be applicable to other types of speciation where geographic isolation seems to be the main mechanism that leads to the splitting of an ancestral lineage (i.e. allopatric speciation). Since we implemented our taxon definition is an independent algorithm (spectral clustering) that does not affect the eco-evolutionary processes of the model, one could modify this algorithm to accommodate other variables (e.g. geographical location of individuals or time since the last branching point). However, such a change to the taxon definition will not impact the eco-evolutionary processes and thus the speciation type captured by the eco-evolutionary model will remain to be a type of non-allopatric speciation, i.e. adaptive speciation (Dieckmann et al., 2004). We have added the following lines to the introduction (lines 51-52) and taxon definition (lines 155-159) parts to better clarify this aspect in the main text.**

… thus the traits are not simply an input or a parameter of the model but a dynamic variable predicted by the model (Klausmeier et al., 2020)…

… This taxon definition is primarily based on the divergence of individuals' traits since they are the main dynamical variables affecting selection, mutation, and competition. However, the spectral clustering algorithm can be modified to accommodate other variables (e.g. geographical location of individuals or time since the last branching point). Furthermore, the taxon definition is independent of the eco-evolutionary processes of the model, thus changing the taxon definition would not affect the model behaviour.

*Mutation: The authors implement a mutation process described as "an intergenerational stochastic variation of trait values". I think the implementation is very reasonable, but I think it may be important (**especially** for geoscience readers) to clarify that the mutation process here is more conceptually complicated than a mutation rate in units base pair/time. Here, the model assumes that mutation affects a specific trait optima, when, most mutations are likely deleterious. I don't think these additional complications need to be added to the model and they are discussed in Irwin (2012), but it would be useful for readers to know if they embark on using the code as the specific understanding of what mutation means as it has implications for the values used for rate of that process.*

**Good point. We have added the following lines to our description of mutation in lines 91-97 of the revised manuscript.**

The mutation process is rather more complicated in reality as we simplify it here and can be defined in its broadest sense as a heritable change in the genome, which can be measured as the number of base pair substitutions per generation and is positively related to the genome size of the organism (Lynch, 2010). Changes in the genotype could lead to phenotypic variation that may be positively or negatively influenced by natural selection, therefore mutations could be silent, advantageous or deleterious (Bromham, 2016). Our model simplifies this genetic–phenotypic complexity by assuming that a change in the genotype (via mutation) directly corresponds to a change in the phenotype (trait), the latter of which is subject to selection and transmitted by uniparentally inherited markers (in the absence of sexual selection) as in Irwin (2012).

*Line 180-181: The authors claim that the model "produces well known patterns of evolutionary branching under fewer 500 generations along a continuous gradient". This claim requires more justification. What exactly is the pattern of branching that is observed and what aspect of it matches "well known" patterns?*

In the context of adaptive dynamics (Metz et al., 1996; Geritz et al., 1998; Dieckmann et al., 2004) and related theoretical frameworks (Abrams et al., 1993; McGill and Brown, 2007; Klausmeier et al., 2020), evolutionary branching refers to the split of a population into two with a widening gap between their average trait values. At the branching point, the resident population coexist with individuals with rare mutations. However, as the competition for resources increases selection leads to these coexisting groups of individuals with different trait values (resident and mutant) to experience opposing directional selection. This divergent drive in their trait values helps to reduce competition and to establish two distinct populations, where each differs in their average trait values. Notice that here we refer to trait, but in the literature, they can also refer as characters or evolutionary stable strategies. These are the well-known patterns we allude to, and that are well illustrated, we believe, for a linear environmental gradient in the seminal paper by Doebeli and Dieckmann (2003). We have added the following lines 206-212 in our revised manuscript to explain these aspects better.

The simple eco-evolutionary model produces patterns of evolutionary branching under fewer than 500 time steps along a continuous environmental gradient. Where evolutionary branching is here referred to as the split of a population with an average trait value into two with a progressive widening gap between their average trait values. The emergence of evolutionary branching along a continuous environmental gradient is a well-known phenomenon captured in eco-evolutionary models that build on the adaptive dynamics theoretical framework (Metz et al., 1996; Geritz et al., 1998; Dieckmann et al., 2004; Klausmeier et al., 2020) and exemplified by the seminal work of Doebeli and Dieckmann (2003).

*Line 190: The local carrying capacity is lowered to reach a similar total abundance of ~400 individuals. Is this done by trial and error or is there a way to a priori decide on these values needed to produce models with similar abundance?*

**Yes, we selected the parameter values manually to reach a similar total abundance. We now added a sensitivity analysis on the maximum number of individual as we vary the parameters $\sigma_f$, $\sigma_u$, $\sigma_d$, $r$, and $K$ (lines 222-223, 431-440 and Figure A1).**

… but see Appendix A for a sensitivity analysis of the effects of selected parameters on the maximum abundance of individuals.

**Appendix A: Sensitivity on the maximum number of individuals to changes in selected parameters**

Due to stochastic and nonlinear relationships of the processes, as well as our definition of the local neighbourhood in the eco-evolutionary model, it is difficult to know a priori how the different processes will interact to produce a maximum number of individuals. Therefore, we perform a sensitivity analysis on the maximum number of individuals, $N_{max}$, to changes in the parameters $\sigma_f$, $\sigma_u$, $\sigma_d$ in relation to changes in radius $r$ and local carrying capacity $K$. We then tested 10 values for the following ranges of $\sigma_f$=[0.2, 2], $\sigma_u$=[0.2, 2], and $\sigma_d$=[10, 100] in relation to changes in 10 values in the range of $K$=[25, 75] and $r$=[25, 75]. We performed 100 simulations for each pair-wise set of parameters for a total of 700 simulations (Figure A1). We observed that the maximum number of individuals will be reached as $r$ decreases and $K$

increases. Also, the maximum number of individuals when K increases and $\sigma_u$ and $\sigma_d$ decreases or when $\sigma_f$ increases. Similarly, the highest $N_{max}$ is reached when $\sigma_u$ and $\sigma_d$ decreases or when $\sigma_f$ increases, in combination with a decrease in the radius of the local neighbourhood $r$.

[Figure]

Figure A1. The maximum number of individuals, $N_{max}$, reached during a simulation given a set of parameters. The model results are calculated using the Example 3.1 setup for a linear environmental gradient. The default parameter values are: $r$=30, $K$=50, $\sigma_f$=0.2, $\sigma_u$=[0.2, 2], and $\sigma_d$=30.

*Temporal scaling: The mechanism for linking the eco-evo component of the model to the LEM in time was not clear to me until the very end of the manuscript (Lines 323-331). It would be helpful to discuss how they are linked in time (e.g. one generation = one LEM time step) earlier in the manuscript, probably in section 3.2.*

**Good point. We have now mentioned the following in section 3.2 (lines 258-262) of the revised manuscript.**

For the coupled execution of the eco-evolutionary model into the LEM we have to assume that one generation time is equal to one time step of LEM. This of course can lead to unrealistic generation times that exceeds the average lifespan of organisms. Therefore, careful consideration of the parameters is required when AdaScape is coupled with FastScape since one generation would then represents the temporal aggregation of numerous real generations (see section 5 for discussion on scaling).

*Effect of competition: One of the main results from the study is that taxon richness is much higher with trait-mediated competition as compared to without trait-mediated competition. In fact, this difference seems to be greater than the comparison between a static versus dynamic landscape. Does this mean that in the face of*

*selection and competition the landscape dynamics are not very important for generating biodiversity? I think the manuscript would be improved with a deeper consideration of the implications of their results for understanding the importance of landscape dynamics.*

**In light of the reviewer's comments, we now see how our results could be interpreted to imply that landscape dynamics are not important in the build-up of biodiversity. However, we do think that both ecological and evolutionary dynamics are influenced by changes in the environment. Nevertheless, our implementation of competition does promote the build-up of biodiversity, but it does require a type of heterogeneity in the environment to generate variability and particularly trait divergence. To better illustrate this, we performed a model run as Example 3.1 for the case with competition ($\sigma_u$) and manipulate the environment – optimal trait relationship ($\alpha$). We set $\alpha$ from the default value of 0.95 to 0, the latter meaning that there is no relationship between the trait and the environmental field. We then run this model setup under an environment with a linear environmental gradient (as in Example 3.1) and under a constant environment centred at the middle elevation. We added the new figure as well as its interpretation in Appendix B (lines 441-451 and Figure B1). Also, we included a short discussion about the role of competition in the build-up of diversity (lines 399-413).**

**6 Competition as a driver of diversity**

Competition for resources is an important ecological process (Tilman, 1982; Chesson, 2000) that can lead to the divergence of traits and consequently promote biodiversity (Pfennig and Pfennig, 2009). Examples of both interspecific (e.g.: Grant and Grant, 2006; Grainger et al., 2021) and intraspecific (e.g.: Bolnick, 2001; Calsbeek and Cox, 2010) competition, are known to leave an imprint on the traits under selection. Therefore, ecological processes have the potential to alter the outcome of evolution. Increasing interest in the past decades have been in documenting cases where ecological dynamics and evolutionary dynamics show reciprocal interactions (Fussmann et al., 2007; Schoener, 2011; Govaert et al., 2019), thus leading to a recurring call to integrate the distinct disciplines of ecology and evolution, as recently point out by Loreau et al. (2023). This becomes particularly relevant knowing that both ecological and evolutionary dynamics can operate at the same pace (Fussmann et al., 2007; Schoener, 2011; Govaert et al., 2019) and can be influenced by rapid changes of the environment, for example as the climate changes (Parmesan, 2006; Loreau et al., 2023). Hence, the difficulty of understanding the tangled relationships between the biotic and abiotic environment with the ecological and evolutionary responses of organisms. Our model, although aiming at capturing the essential eco-evolutionary processes, simplifies much of the organism-organism and organism-environment feedback. Nevertheless, the results support the general view that competition is an important process that promotes the build-up of taxon diversity.

**Appendix A2: Trait-mediated competition, optimal trait-environment relationship and environmental heterogeneity**

The effects of competition in our model depend on the relationship between the environmental field and optimal trait, as well as on the heterogeneity in the environment. To better illustrate this, in Figure B1 we performed a model run as in Example 3.1 for the case with competition ($\sigma_u$), where we set the environment – optimal trait relationship ($\alpha$) from the default value 0.95 to 0, the latter meaning that there is no relationship between the trait and

the environmental field. We then run this model setup under an environment with a linear environmental gradient (as in Example 3.1) and under no environmental gradient, i.e. a constant environmental field centred at a middle elevation. We observe no build-up of taxon diversity when there is no environment–optimal trait relationship ($\alpha = 0$) without (Figure B1 A-D) and with (Figure B1 E-H) linear environmental gradient, and when $\alpha > 0$ with a linear environmental gradient (Figure B1 I-L). Trait-mediated competition thus only promotes diversity when a trait environmental relationship exists and when the simulation occurs in a heterogeneous environment (as in Figure 3 F-J).

[Figure]

Figure B1. Effects of the environment - optimal trait relationship ($\alpha$) and competition on the build-up of taxon diversity. The numerical experiments show the results of simulations with 1) no environmental gradient (same elevation throughout the landscape) and no environment - trait relationship $\alpha$ (A-D), 2) linear environmental gradient (as in Figure 3) and no environment - trait relationship (E-H), and 3) no environmental gradient and a positive environment - trait relationship $\alpha =$ 0.95. The other parameter values are set as in Figure 3 for the case with the trait-mediated competition.

*Figure 4G: I think there are a grey line and a thin dashed line along the x-axis that can't be seen easily. I suggest dropping the axis down slightly below 0 so the lines can be seen. I was confused about why there were only wo lines in Figure 4G when Figure 4N has three (min, mean, max).*

**Corrected.**

*Figure 5: It is hard to see the 2D histogram, can the lines be darkened?*

**We tried to use other colormaps but the contrast is still not very visible. Therefore, we reduce the number of bins along the y-axis to make the density more visible.**

*Technical Corrections (grammar, wording, etc):*

*Line 3: "an ideal tool" --> "ideal tools" (referring to models which is plural)*

**Corrected.**

*Line 12: "observed" --> "observe" (present tense)*

**Corrected.**

*Line 25: "challenge to study" --> "challenge of studying"*

**Corrected. See line 26.**

*Line 25: I think "recalled" is not the right word, perhaps "utilized"?*

**Corrected. See line 26.**

*Line 30: The final sentence of the paragraph here is a bit hard to understand but is crucial for distinguishing the model here from previous efforts where eco-evo models run on static landscapes, suggest rephrasing to better emphasize the difference.*

**Thanks for pointing this out. We have now rephrased the final parts of this paragraph also in light of yours and RC2 suggestions (lines 28-35).**

Currently, these types of models are known as "population-based spatially explicit Mechanistic Eco-Evolutionary Models" or MEEMs (Hagen, 2022) and prominent examples include Rangel et al. (2018) and Hagen et al. (2021). In these models, the main emphasis is on how species interact and evolve, in a grid-based environment, where environmental fields (e.g. topography, temperature, and precipitation) are representations of past or present features computed a priori. These landscape representations are provided by other models, such as global paleo-elevation reconstructions (e.g. as in Hagen et al., 2021) or after reanalysis of the output of other models (e.g. as in Rangel et al., 2018). These MEEM tools offer flexibility in the inputs and detail treatment of the ecological and evolutionary processes, however, they provide little control over the mechanisms that generate climate and landforms.

*Line 31: You aren't generating landforms themselves, suggest rephrasing as "Generating landforms in simulations..." or "Simulating landforms..." or "Understanding how landforms are generated..."*

**Corrected. See line 36.**

*Line 51: "which is" --> "which are" (subject is plural)*

**Corrected. See line 57.**

*Line 182: suggest deleting the word "exemplary" which would imply two really stand-out and impressive results (where I think the authors mean two typical examples of results). Same thing on line 203.*

**Corrected. See line 213.**

*Line 210: "the simulations ultimately reach equivalent mean, maximum, and minimum values" (of what?)*

**Thanks. We have rephrased the sentence (lines 243-244).**

The simulations for static and dynamic landscapes reach equivalent mean, maximum, and minimum values of elevation and precipitation.

*Line 218: "To large differences" ....unsure what that means, typo?*

**Thanks, we were missing a word (line 249-250).**

To avoid large differences...

*Line 253: "as the rate of uplift is getting slower" --> "as the rate of uplift slows"*

**Corrected. See line 290.**

*Line 257: "show also" --> "also show"*

**Corrected. See line 294.**

*Line 269: "varies" --> "vary"*

**Corrected. See line 305.**

**RC2**

*1. General comments*

*The manuscript presents a landscape evolution and individual based model integration. This is not new per se, but the integration the authors provide is a very valuable asset for studying the links between geo and bio dynamics. Throughout the manuscript the authors give the impression that the other previous models (e.g. Rangel's and Hagen's) cannot be used to study this link. The authors should make clear that this is not the case, while reinforcing the advantages of their tool since the integration of these two different models is provided in one software. Additionally, further description and discussion on assumptions taken, especially for the evolutionary component, should be clarified (e.g. individual based, speciation, taxon definition, mutation, gene flow, landscape residence, "generations vs timesteps" and runtime). Furthermore, deeper discussions and clarifications on the limitations such as computational time and limited biological processes implementation and uncertainties related to individual based modelling approach and their taxon definition and mutation process should be added. Deep-time eco-evolutionary models have been harshly criticized due to their large number of parameters leading to identifiability and structural error problems. I would like to see the authors defend this further and justify the advantages of adding these two models together while increasing inevitably the number of parameters. Specific comment suggestions point out what could be changed and what would benefit from further clarification.*

**Thank you for raising these valuable issues. Regarding your first point about Rangel's and Hagen's works, it was not our intention to give the impression that their valuable contributions cannot be used to study the link between geo and bio dynamics.  We have clarified this below in your specific comment. Regarding the points about further description and discussion about the processes mentioned, please refer to our responses to your specific comments below and similar comments raised by reviewer 1 above. Regarding your point about defending the effort to couple components of the Earth system in deep-time simulations, we have now added a few lines on the approach we took to build our model (lines 360-366) and a potential advantage of coupling an eco-evolutionary model with a landscape evolution model (lines 423-428).**

Any model at its best is a surrogate of nature that we can use to test our understanding of a system. We decided to build on established theoretical frameworks to study the coupled eco-evolutionary dynamics (Metz et al., 1996; Geritz et al., 1998; Dieckmann et al., 2004; McGill and Brown, 2007; Klausmeier et al., 2020) and landscape evolution (Whipple, 2004; Tucker and Hancock, 2010; Lague, 2014). These theoretical frameworks have had many applications over past decades with more detailed descriptions of processes as we have shown here, we thus use only their essential processes to illustrate how bio- and geo- components of the Earth

system can be coupled into a single modelling framework while keeping the number of parameters and processes to a minimum.

… At the moment AdaScape mainly considers the effect that the environment has on the biota, however, feedbacks between these components can be further investigated. For example, by linking the effects that organisms such as plants have on the hydrological cycle, which are known to dampen the discharge variability (Rossi et al., 2016) and therefore the erosional efficiency of rivers where erosional thresholds exist (Deal et al., 2018; Braun and Deal, 2023). This would require to investigate and test relationships between organismal traits with erosional processes in the landscape, which offer a potential venue to further integrate various components of the Earth system into a single modelling framework.

*2. Specific comments*

*I found the use of elevation, rather than a proxy of temperature rather unusual. For example, do you expect elevation to act similarly on closely related groups and higher and lower latitudes? Macro-ecologists would benefit from a more direct proxy for biological processes and I wonder if this could be implemented or at least discussed upon.*

**We use elevation since this is the main output of the LEM and it could be used as a proxy for temperature assuming lapse rates of around 6.5 °C/km and a base temperature at sea level. We have added the following lines to clarify this aspect in lines 183-186.**

While elevation is the main output of the LEM, this environmental field could be used as proxy for temperature. This would require further assumptions for example, that temperature decreases with elevation around 6.5°C/km (Minder et al., 2010) and a given baseline temperature at sea level, which could be constant or change over time and taken from climatic paleo-reconstructions.

*The authors present an individual based model of biodiversity to investigate deep-time questions, which is in strong difference with most current approaches investigating deep-time which use population based models to tackle geodynamics [https://doi.org/10.1111/ecog.06132]. I wonder what the computational constrains and advantages of this approach are. Specifically, it would be handy for readers to have an idea of computational requirements for a given spatial and temporal extent with some average parameter setting (or at least the spread of CPU time for different scenarios). 10'000 generations in about 80min seem very time consuming to simulate anything global and deep-time where plate tectonics are relevant. Consider providing more information of spatial and temporal extent coverage and resolution as well as possible processes that can be considered*

**Thanks for pointing us out to the review of Hagen on MEEMs. We have added a further discussion and comparison of individual-based modelling approaches with MEEMs following your suggestions here and on the following comment. Regarding the computational requirements, the simulations using our framework can be run on a common desktop/laptop. Albeit, for efficiency we have computed our simulations in our Linux cluster. We have added further details of the temporal and spatial extent of the execution in lines 377-381.**

For all simulations, the spatial extent consists of an area of 100 km by 100 km, which is divided in a regular grid of 100 by 100 points. The simulations were executed for 10 Myr with variable time steps, as explained in figure 9. Albeit we run these simulations in a 176-core (Intel Xeon 2.10Ghz) Linux cluster, the execution of FastScape and AdaScape do not require any highperformance computing facilities and can be executed in any modern desktop or laptop computer where Python can be installed with similar performance as shown in figure 9.

*Line 25: "In these types of models (e.g.: Rangel et al., 2018; Hagen et al., 2021a), individuals or species interact and evolve, generally, in a grid-based environment, where environmental fields (e.g. topography, temperature, and precipitation) are static representations of past or present features. Consequently, this offers little control over the processes that generate landforms while emphasising the ecological and evolutionary drivers of diversity." This affirmation is wrong in two ways. First, both models consider dynamic environments, with the only difference being that input landscapes are generated prior to the biological model run. This can be a problem for pipelining multiple LEM parameters, but still gives flexibility in choosing or inputting landscapes. I see the biggest difference in these methods that the landscapes are created on the go by your toolbox, or did I get this wrong? This also has consequences that should be emphasized. Secondly, these two models do not consider individuals or species explicitly. These models are population based in order to cope with the large computing requirements when considering deep-time.*

**Thanks for your clarifications, we now see how our description of static vs. dynamic environmental fields in the context of Rangel's and Hagen's work could lead to misinterpretations. We have now revised the relevant parts in the introduction as mentioned to a related comment to reviewer 1 above (lines 28-35) and in the discussion to better contextualized the comparison of our work from those of Rangel's and Hagen's following your suggestions (lines 338-351).**

…in what is known as "population-based spatially explicit mechanistic eco-evolutionary models" or MEEMs for short (Hagen, 2022). Our model differs from MEEMs in that we follow an individual-based (IBM or agent-based ABM) modelling approach, which in comparison to population-based models offers greater flexibility into the processes considered and how the organisms interact among themselves and with the environment (Levin, 1998; Railsback, 2001; DeAngelis and Mooij, 2005; Grimm et al., 2005). IBMs thus account for low-level variability that can be scaled up to higher hierarchical levels, producing emergent properties that cannot be predicted by the properties of individuals or their interactions with the environment alone (Levin, 1998; Railsback, 2001; DeAngelis and Mooij, 2005; Grimm et al., 2005). Nevertheless, accounting for individual-level variability, particularly as observed in nature, can be impractical and computationally demanding. Hence, population-based models, such as MEEMs are a more computationally efficient option (Hagen, 2022). In addition, MEEMs such as Rangel et al. (2018) and Hagen et al. (2021) do not compute the landscape dynamics but use a priori calculated environmental fields. This approach allows them a more flexible and efficient way to upscale the computations. However, this comes at the cost of not controlling the relevant processes that lead to the building of a landform or climate as in our coupled eco-evolutionary and landscape evolution model.

*I was confused at fist on how generation time and discreate time-steps are considered? The authors hint on disadvantages of discrete models, but end up using the same method, sometimes referring to the discretization of input and processes as time steps or generation time. I recommend defining this more clearly and only using time steps. 10 kyr is of an unheard generation time, and using generation time specific for AdaScape would only cause confusion.*

**Thanks for pointing this out and as also suggested in similar comments by reviewer 1, we now mention the time differences between LEM and *AdaScape* more clearly early-on in the text (lines 258-262) and revised throughout the text to mainly refer to time steps.**

*Another aspect that was not clear is: How does the taxon definition link with gene flow and geographic distance, or is this process not considered at all?*

**As explained to a similar point to reviewer 1 about taxon definition, our taxon definition does not consider geographic distance and barriers to gene flow, it's based on the divergence of traits between groups of individuals. Please see our response to reviewer 1 above and the modified text in lines 51-52 and 155-159.**

*Are you considering landscape resistance? I have trouble in putting dispersal as an evolutionary process. Can you explain better how dispersal is working?*

**At the moment we do not consider any type of landscape resistance to the movement of organisms. We have added the following to the description of dispersal in lines 101-103.**

In other words, dispersal describes the random movement of individuals and their traits through the landscape, where the new location of individuals at *t+1* depends on the location of individuals at time *t* and $\sigma_d$.

*I missed some more explanation about the local neighborhood. Does it also influence trait homogenization? Do you consider this at all?*

**We have extended the description of the local neighbourhood (lines 114-116). Regarding trait homogenization, no we do not consider a specific process for this.**

The local neighbourhood is the area around each individual centred at its location $l_{i,\bullet}$ and with an extend determined by $r$, and can be thought of as the area where local competition for resources take place. The parameter, $\sigma_u$...

*How does the spatial component of the competition work? Seems like it can jump sights as presented on Figure 1 top right.*

**The spatial extent of the competition is determined by the local neighbourhood, in the sense that only the individuals inside that local neighbourhood are considered to be competing for the same resource. See response to previous comment.**

*I did not understand if you have any reproductive isolation? When things diverge on traits but randomly get similar below the threshold, can they again become the same species? Can you please clarify this?*

**At the moment our model does not consider reproductive isolation. Regarding the taxa merging again, we avoided this in our taxon definition algorithm. The traits of different taxa can be similar, but once a taxon has been split it cannot merge again.**

*3. Technical corrections*

*Lines 8-9: "Particularly, we investigate how changes in tectonic uplift rate and ecoevolutionary parameters (i.e. competition, dispersal, and mutation) influence the temporal and spatial patterns of biodiversity". I wonder why you did not add speciation here, since you focus on deep-time and mention speciation as the only process on Line 39: "Here we present AdaScape, a coupled speciation and landscape evolution model".*

**We were avoiding repetitions, but for clarity in light of the reviewer comments we have include speciation in lines 8-10.**

Particularly, we investigate how changes in tectonic uplift rate and eco-evolutionary parameters (i.e. competition, dispersal, and mutation) influence speciation and thus the temporal and spatial patterns of biodiversity.

*Line 115. I could not follow what you meant by: "all individuals have been updated".*

**Here by update we refer to the point during the execution of the simulation where the internal state of individuals or agents is changed following the established rules or procedures, as commonly done for individual-based models (Railsback and Grimm, 2019). In our model, these rules/procedures are fitness, mutation, dispersal, and competition.**

*Line 123. Something is off here "then branching does not occur all individuals will be assigned to a single taxon-id".*

**Thanks, we were missing a word (line 140).**

… then branching does not occur and all individuals will be assigned to a single taxon-id …

*Line 127-128. You can reword this to "only considering binary splits", or "artificially transforming polytomy into simple splits". This could resonate better with biologists.*

**Thanks, we have added the following (line 143).**

… with this we are avoiding polytomies by considering only binary splits.

*Line 130. Close ")".*

**Corrected. See line 147.**

*Line 326-327. "The most adequate scaling is to consider that the mutation and dispersal rate need to be scaled by the square root of the number of real generations in a LEM time step." Citation for the scaling proportion is missing. Consider only mentioning time step throughout since these are not only exclusive to LEM.*

**We are proposing this scaling. In light of your comments, we now realised that we were too brief in our explanations. Therefore, we have extended those lines to better explain the scaling, how it can be used to interpret model parameter values for mutation and dispersal, and compared to measured values (lines 258-262, 382-394).**

When AdaScape is coupled with FastScape, the time between generations is defined as the time step of the LEM (i.e. 10 kyr for the example simulations shown in section 3.2). Therefore, a generation in AdaScape would represent the temporal aggregation of numerous real generations. In this context, a scaling of the eco-evolutionary parameters related to mutation ($p_m$, $\sigma_m$) and dispersal ($\sigma_d$) must be considered. The simplest way is to scale the mutation and dispersal parameters by the square root of the number of real generations in a LEM time step. Therefore, the parameter $p$ used in a given simulation with a time step $\Delta t$ should be scaled for comparison with measured values, $p'$, by the following relationship: $p' \sim p \cdot \sqrt{t_G / \Delta t}$,

where $t_G$ is the average lifespan of a generation, $p$ is one of the mutation or dispersal parameters, and $p'$ is the new scaled parameter. Caveats of this scaling are that the parameter value is not directly constrained by observations and that obviously, this parameter should not be too large in a simulation to avoid, for example, that dispersal exceeds the length of the simulated area or that all individuals mutate and vary the trait value broadly in a single time step. Particularly, the latter goes back to the assumptions of earlier formulations of adaptive dynamics, stating that the range of validity of these types of eco-evolutionary models stay as long as mutation processes were rare and the trait variability with respect to their ancestor was small, to assure that evolutionary dynamics were slower than changes in population densities (Abrams, 2001; Klausmeier et al., 2020).

*Line 328-331. You could add some further discussion here on the comparison of individual and population based models, or just provide citations to this other body of work.*

**We have expanded our discussion on the comparisons between population and individual-based models. See response to related comments above and lines 3387-351.**

*I could not follow on section 2.1 how individual traits change, or if they are only selected. If only selected, I would rather call this an ecological, rather than an evolutionary process, making mutation the first evolutionary process (Line 78). The same applies to dispersal.*

**The trait of individuals is change via mutation. We have revised the description of mutation and dispersal in response to yours and reviewer 1 suggestions see lines 91-97 and 101-103.**

*Could you justify the selection of sigma_u=2? I would expect a value of 1 for the experiment where trait proximity is irrelevant for competition.*

**The competition as describe in equation 4 is a Gaussian function, which approximate the effective number of individuals based on their similarity in trait values for given $\sigma_u$. Therefore, to count for a discrete number of individuals is that we choose $\sigma_u$ value greater 1, to make sure that as close as possible all individuals are counted like in the case of without trait-mediated competition.**

[revised manuscript text omitted]